# Warm proglacial lake temperatures and thermal undercutting enhances rapid retreat of an Arctic glacier

Adrian Dye[1, 2], Robert Bryant [3], Francesca Falcini[4], Joseph Mallalieu[5], Miles Dimbleby[2], Michael Beckwith[6], David Rippin[1] and Nina Kirchner[7]

[1] Environment Department, University of York, York, YO10 5NG, United Kingdom
2 Environment Cluster, Teesside University, Middlesbrough, TS1 3BX, United Kingdom
3 Geography Department, University of Sheffield, Sheffield, S3 7ND, United Kingdom
4 Alfred-Wegener-Institut Helmholtz-Zentrum für Polar und Meeresforschung, Bremerhaven, Germany
5 School of Geography, Earth and Environmental Science, University of Birmingham, Birmingham, B15 2TT, United Kingdom
6 Independent Researcher
7 Department of Physical Geography, and Tarfala Research Station, Stockholm University, Stockholm, 106 91, Sweden

*Correspondence to*: A. Dye (a.dye@tees.ac.uk)

**Abstract.**

Determining the characteristics of Arctic proglacial lakes is essential for understanding their current and future influence on glacier mass loss, capacity as a carbon sink and the associated impacts for downstream hydrology and ecology. Field observations of how proglacial lake properties influence rates of glacier mass loss remain sparse, yet are increasingly critical for the accurate projection of lake-terminating glacier responses to warming air and lake temperatures, particularly in high-latitude Scandinavia under the influence of Arctic amplification. Here we combine satellite and field observations of Kaskasapakte Glacier (KG) (a lake-terminating glacier in Arctic Sweden) to reveal the interplay between lake parameters and glacier mass loss from 2008-2019. We present the first field evidence of warmer than expected water temperatures (>4 ºC at the ice front) at a Scandinavian proglacial lake and illustrate how these drove rapid thermo-erosional undercutting and calving at the terminus, with width averaged retreat rates of up to 25 m per year and frontal ablation accounting for ~30% of glacier volume loss between 2015 and 2019.

Keywords;

Proglacial Lake, Ice-Marginal Lake, Calving, Arctic Sweden, Arctic Amplification, Glacier Retreat, Sonar

## 1 Introduction

Where glacier termini are in contact with proglacial lakes, the latter have been shown to accelerate glacier mass loss rates through thermal and mechanical processes at sites in Alaska (Boyce et al., 2007), Patagonia (Skvarca et al., 1995; Minowa et al., 2017), Nepal (King et al., 2017), Greenland (Mallalieu et al., 2021), the Russian Arctic (Carr et al., 2014) and New Zealand (Warren and Kirkbride, 1998; Röhl et al., 2006). Whilst previous studies have reported small proglacial lakes to be a uniform

1°C (Truffer and Motyka, 2016), proglacial lake temperatures of >4 °C have been reported in Nepal, New Zealand, Patagonia and Arctic Sweden (Kirkbride and Warren 2003; Sugiyama et al., 2016; Watson et al., 2020; Dye et al., 2021). High

subaqueous melt rates remove mass from the glacier terminus and cause thermal undercutting (producing thermal notches) that promotes iceberg calving through failure of overhanging subaerial ice cliffs (Iken, 1977; Warren and Kirkbride, 2003; Röhl et al., 2006). Proglacial lake temperatures have been found to control seasonal ice front position in Patagonia and terminus morphology in New Zealand, where rapid (0.65 cm d$^{-1}$) thermal notch development has led to substantial undercutting of glacier termini (Minowa et al., 2017; Röhl et al, 2006). Such thermal undercutting alters the profile and stress balance at the

ice front, as support is removed from lower down the subaerial ice front (producing a 'top heavy' profile) so stresses increase within the ice, which may either fracture above the overhang (calving icebergs) or develop crevasses parallel to the ice front (Iken, 1977). Therefore, the glacier terminus position in glacial lakes is a delicate balance between dynamic ice fluxes towards the terminus and ablation processes at the ice-water interface, both of which respond sensitively to changes in climate and/or changes in lake characteristics (e.g. water temperature) (Minowna et al., 2017).

Arctic amplification of climate change has increased air temperatures in the Arctic by four times the rate of Northern Hemisphere warming (Rantanen et al., 2022). Consequently, constraining the temperature of Arctic proglacial lakes is essential to understand their current and future influence on glacier retreat rates and associated impacts on downstream temperatures and ecology (Richards et al., 2012; Fellman et al., 2014; Kim et al., 2019; Woolway et al., 2019; Yiou et al., 2020), and their capacity to act as a carbon sink, particularly where suspended sediment concentrations from glacial meltwater are high

(Sugiyama et al., 2016; St Pierre et al., 2019; Watson et al., 2020). To date, there have been few process based field studies at proglacial lakes in the Arctic, despite being associated with enhanced glacier retreat rates around the Greenland Ice sheet and Novaya Zemlya (MacIntyre et al., 2009; Carr et al., 2017; Carrivick and Quincey 2014; Carrivick et al., 2017; Mallalieu et al., 2021; Carrivick et al., 2022). This study aims to investigate the relationship between glacial lake temperatures and glacier retreat, (on a decadal and seasonal timescale) at a previously unstudied Arctic glacier through remote sensing and in situ

measurements. Here, we present the first recorded in-situ proglacial lake temperature record from the front of an actively calving glacier in the Scandinavian Arctic, combined with analysis of high-frequency time-lapse imagery of calving events, sonar surveys of lake bathymetry and the sub-aqueous ice-front, multitemporal satellite imagery and digital elevation models (DEMs). Combined, this represents a detailed investigation into the influence of proglacial lake temperatures on mass loss of an Arctic glacier. The objectives of this study are:

1. Measure terminus recession between 2008-2019 from remote sensing and investigate its relationship to proglacial lake bathymetry.

2. Document changes in the subaqueous (2019) and subaerial (2017-2019) geometry of the glacier terminus.

3. Classify calving mechanisms and investigate environmental drivers of calving throughout the 2019 melt season.

4. Assess the influence of lake water temperature and climate on terminus evolution at Kaskasapakte Glacier.

## 2. Study Area

Kaskasapakte Glacier (KG) is ~2 km long and flows northeast from two subsidiary corries (located below ~500 m headwalls, with peaks of ~2,000 m asl to the east, south and west) into the main trunk; currently terminating in a calving front in an unnamed proglacial lake (now referred to as KGL) with some latitudinal supraglacial debris bands near the terminus (Fig. 1 and see Supplementary for images). KGL is situated at 1,100 m asl and was 670 m long, with a surface area of 0.13 km$^2$ in August 2014; it has an outlet at the northernmost point and freezes over during winter (Dye et al., 2022). KGL has expanded since ~1916 CE as the glacier has retreated from its Little Ice Age maximum position at the terminal moraine (Karlen, 1973). KG had the largest retreat (126m) of any glacier in the Kebnekaise between 2010 and 2018 (Dye et al., 2022). At the start of fieldwork in 2017 the glacier termini was observed to have two englacial conduits at lake level, that became active most afternoons (as demonstrated by export of icebergs out from caves). The nearest weather station is 5 km south of the study site at Tarfala Research Station (1,135 m asl; 67.9124º N, 18.6101º E) where the mean annual air temperature (MAAT) from 1965-2011 was -3.5 ±0.9 ºC, but the MAAT has risen to -2.2 ºC between 2012 and 2022; furthermore the average air temperature for July has increased from 7.0 ºC (1965 to 1994) to 8.09 ºC (1990 to 2019) (Jonsell et al., 2013; Dye et al., 2021; Kirchner et al; 2023). Recently the area has also experienced pronounced heatwaves (month long), with August 2014 and July 2018 being 5.4 ºC and 5.6 ºC above the long term average (Dye et al., 2022).

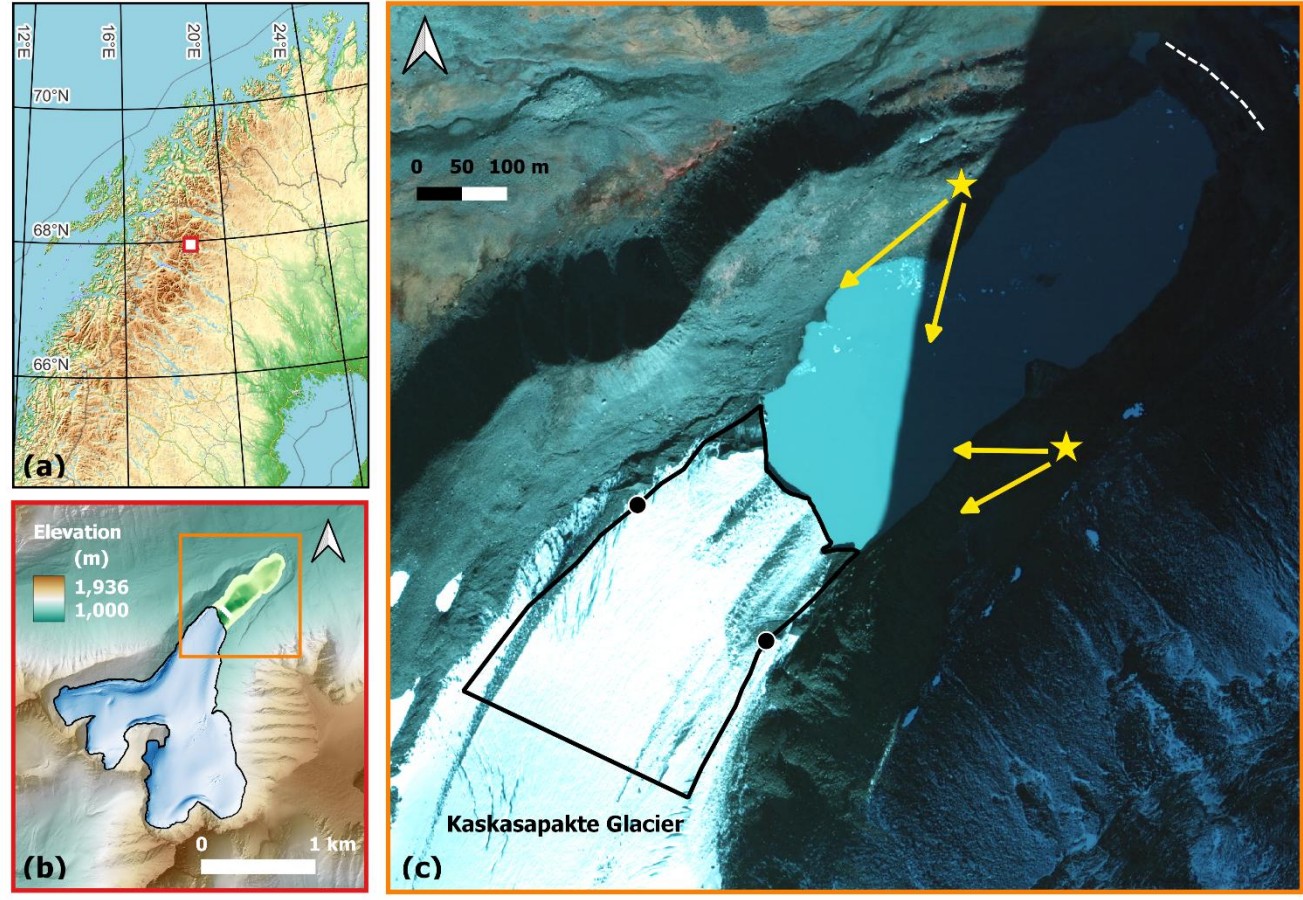

**Figure 1. (a) Location of Kaskasapakte glacier (67.95721ºN, 18.56109ºE) in Arctic Sweden (red box). (b) Digital Elevation Model (2m) and digitised glacier margin (black outline) from Lantmateriet Lidar survey August 2015 with bathymetry data from field surveys (c) Lantmateriet (0.5 m; 2008) Orthophoto with terminus polygon (black line) and width reference points (black circles). Yellow stars = timelapse camera positions and orientation. White dashed line = Little Ice Age maximum terminal moraine (in shadow top right corner) (Karlen, 1973).**

## 3 Methods

A series of different fieldwork surveys were carried out in conjunction with remote sensing in order to assess how the glacier terminus geometry has changed in relation to changes in climate and lake characteristics (Table 1). Two periods of fieldwork were conducted (between 23rd July-4th August 2017 and 29th July-10th August 2019) at KGL, in the Kebnekaise massif (Arctic Sweden), into which Kaskasapakte Glacier (KG) terminates.

**Table 1 - showing data set, objective, timescale and years associated with each method.**

| Data | Satellite Imagery | Sonar | Side-view sonar | DEMs | Timelapse | Thermistors | Met Data |
|---|---|---|---|---|---|---|---|
| **Objective/ target** | terminus recession | bathymetry | subaqeos ice front | volume change | calving mechanisms | lake temperature | Meteorological conditions |
| **Timescale** | interannual | annual | annual | interannual | 3 hourly | hourly | hourly |
| **Dates/Years** | 2008 - 2019 | 2019, 2022 | 2019 | 2015, 2019 | 2017, 2019 | 2019 | 2019 |

## 3.1 Width averaged terminus position change from remote sensing (2008 – 2019)

Orthophoto imagery (September 2008) from Swedish Lantmateriet was used as a baseline for assessing terminus geometry changes. Cloud and snow free RapidEye multispectral satellite images (5 m spatial resolution) of KG were downloaded for 16th August 2010, 14th August 2012, 20th July 2014, 24th August 2016, and 27th July 2018. To account for variations in visible glacier termini geometry (due to changes in debris cover) a consistent reference point (black circles; Fig.1) was selected on either side of the glacier where the width was kept consistent for each year between 2008 and 2018 (following the curvilinear box method; Lea et al., 2014) (Fig. 1). The glacier terminus area was then mapped downstream of these consistent width reference points to create terminus polygons for 2008, 2010, 2012, 2014, 2016 and 2018. Each terminus area was then subtracted from the preceding terminus area iteratively to calculate the area of ice lost over that period, and divided by the width between the polygon reference points (black circles; Fig. 1). This was then divided by the decimal year value (number of days divided by 365) for the period in each image acquisition. Thereby giving a width averaged retreat rate for the number of days during the period between image capture dates, thus accounting for the offset between image capture dates between years.

## 3.2 Lake bathymetry and subaqueous terminus geometry from sonar mapping

KGL has two distinct basins that were surveyed in summer 2022 with a Garmin echoMAP CHIRP 45cv sonar (with 5 Hz GPS/GLONASS and thermistor for temperature calibration) mounted on an Autonomous Surface Vessel (ASV). During the field season in 2022 downward looking sonar scans (at 200 kHz) were run in a grid near the ice front in conjunction with two survey lines along the lake axis from an inflatable kayak (up to 200m from the ice front), from which depth points were extracted (at 30 second intervals) and interpolated in QGIS (IDW algorithm within August 2019 lake polygon) to create the bathymetry map (see Supplementary S1). A series of side view sonar surveys were conducted from the ASV in 2019 near the ice front to map the glacier's subaqueous geometry, with the sonar transducer mounted side looking to collect sonar images using the CHIRP ClearVu setting (200 kHz); these images were viewed and trimmed in SonarTRX software before being exported to QGIS. Previous studies have reported vertical uncertainties of +/- 5.9m at 50m depth with Garmin Fishfinder sonars in comparison to high resolution Odom echo sounder (Purdie et al., 2016).

### 3.3 Terminus geometry change

### 3.3.1 Glacier volume change 2015 to 2019

We calculate 'static' glacier volume change for geometric differences between glacier surfaces from 2015 and 2019 for the lower part of KG, as insufficient velocity data prevented dynamic processes being incorporated. Eighty overlapping images of the glacier terminus were recorded on 3$^{rd}$ August 2019 from vantage points spaced along the lateral and terminal LIA moraine crests using a handheld Canon EOS 70D 20-megapixel digital single lens reflex (dSLR) camera. A digital elevation model (DEM) of the glacier terminus on 3$^{rd}$ August 2019 was generated by processing the images using Structure-from-Motion (SfM) techniques in Agisoft Metashape. The SfM DEM has a 0.2 m pixel and was georeferenced using ground control points (GCPs) of stable landscape features either side of the glacier terminus (large boulders, exposed bedrock; see Supplementary S1) recorded with a Trimble dGPS and reprojected to SWEREF 99 TM (EPSG 3006) (following Mallalieu et al. 2017). Elevation model uncertainty relative to the GCPs was; $z = 0.05$ m, $x = 0.04$ m $y = 0.04$ m (Wilkinson et al., 2016). To calculate volume change of the glacier the 2019 DSM (SfM) was subtracted from the Swedish Lantmateriet Lidar derived Digital Elevation Model (2 m pixel, z residual error of c. 1m; EPSG 3006; Mannerfelt, 2022) in QGIS to generate a DEM of Difference. When comparing the 2015 Lidar DEM and 2019 SfM DEM, an elevation RMSE of 0.52 m between the two dates was calculated for known stable areas of the moraine. This allowed evaluation of 'static' change in terminus geometry using the following equation to account for changes in the glacier (*KG*) subaerial surface and subaqueous surface below the lake (*KGL*) level;

$$\text{Volume change } (V_c) = (KG_{surface\ 2015} - KG_{surface\ 2019}) + (KG_{surface\ 2015} - KGL_{level\ 2019}) + (KGL_{level\ 2019} - KGL_{bed\ 2019})$$

In order to partition volume loss between surface lowering and terminus recession, 3 separate components were calculated. Changes in surface elevation of KG between 2015-2019 was calculated by subtracting the August 2019 SfM DEM from the 2015 Swedish Lantmateriet Lidar derived Digital Elevation Model (2 m) for the 2019 polygon extent (See section 4.3). Beyond the 2019 polygon extent the 2019 KGL level was subtracted from the 2015 KG surface to give the volume changes between the 2019 and 2015 DEMs down to lake level (with a vertical subaqueous ice front assumed in both years). The ice proximal bathymetry was subtracted from the 2019 KGL level (using QGIS volume calculation tool) to calculate the subaqueous ice volume lost due to terminus recession between 2015 and 2019 (see Fig. 4 for extent). Thus enabling the volume changes to be calculated for a. volume lost for KG surface lowering and b. KG volume lost (including subaqueous and subaerial) from terminus retreat between 2015 and 2019.

### 3.3.2 Seasonal glacier terminus subaerial geometry changes (2017 and 2019)

Changes in glacier terminus geometry were monitored over a two-week period from 23$^{rd}$ July to 4$^{th}$ August 2017 using an LtL Acorn 5210a 12-megapixel time-lapse camera stationed on the lateral moraine above the eastern shore of KGL and programmed to record pictures at one-minute intervals (Fig.1). In 2019 two cameras (same model) were stationed on the western and eastern KGL shores, and programmed to record images of the terminus every three-hours from 5$^{th}$ August to 19$^{th}$

September 2019 (Fig.1). Subaerial calving events throughout the 2019 melt season were identified in the time-lapse imagery and categorised into seven types; reflecting the primary mechanism and style of calving (ice fall; sheet collapse; stack topple; waterline; subaqueous; roof/arch collapse; and unknown) following the classification of How et al. (2019). Of these calving mechanisms, three are directly associated with undercutting of the subaerial ice cliff at the water line (sheet collapse, waterline and roof/arch collapse) and the remaining mechanisms typically associated with outward or buoyant force imbalances at the terminus. For classification of larger undercut features in the subaerial ice front (above the waterline notch); arches are defined as features where depth of the overhang was less than its height , whereas if the depth of the overhang was greater than its height it was defined as a cave.

### 3.4 Lake water and meteorological changes through the 2019 melt season

Summer 2019 lake water temperatures were measured at 1 m depth on a line suspended from the glacier terminus (67.95396 N 18.55955 E) using a HOBO UAA-002-08 pendant to measure hourly temperature ($\pm$ 0.5 $^{\circ}$C) and light (so periods of solar warming from sensor disruption could be identified for quality control). A thermistor was positioned lower down (at 2m depth) parallel to the ice front, but this was removed by a calving event. Further temperature measurements (also HOBO UAA-002-08) were taken at 20m depth on 8[th] August 2019 for 3 hours and at the central position in the lake (see S1) at 5m deep from 5[th] August to 9[th] September; both results are presented in the supplementary section (see Fig.2 for locations). Hourly air temperatures and precipitation data were downloaded for August and September 2019 from Swedish Meteorological and Hydrological Institute (SMHI, smhi.se) which has an Automated Weather Station (AWS) situated in Tarfala (ca. 5 km from glacier front). Precipitation data (from Tarfala Research Stations AWS) was obtained from Swedish Infrastructure for Ecosystem Science (SITES, 2020) .

### 4 Results

### 4.1 Terminus position change 2008-2019 in relation to lake bathymetry

The glacier terminus sustained a similar position at the start of the observational period with minimal width averaged retreat (section 3.1) of 2 m between 2008 to 2010 (1.06 m y$^{-1}$), followed by a slight increase to 4.95 m y$^{-1}$ (2010 to 2012) (Fig. 2). Retreat rates then increased substantially to 11.66 m y$^{-1}$ (2012 to 2014) and peaked at 25.31 m y$^{-1}$ (2014 to 2016), before dropping slightly to 17.77 m y$^{-1}$ (2016 to 2018). During this period the profile of the terminus (in plan view) changed in shape, with some satellite scenes (particularly from 27[th] July 2018 and 3[rd] August 2019) capturing a rather straight profile, whilst others (particularly from 20[th] July 2014) captured a more curvilinear profile with prominent bays (Fig. 2). Whilst these satellite images represent only a snapshot in time, we report them here as there is noticeable variation between 2008 and 2019 that should be considered in the context of lacustrine terminus geometry changes; as they are over time periods that are important to the typical progression of thermal undercutting by a proglacial lake. Particularly given the variability in englacial conduit position on the terminus and subsequent variability in terminus profile.

The KGL has 2 distinct basins, with the ice distal basin being substantively shallower (max depth ~6 m), whereas the ice proximal basin is typically >14 m deep in the centre (max. depth ~20 m) with shallower sections around the western margin (Fig. 2 and Supplementary S1). The bathymetry survey revealed that central parts of the terminus remained in relatively deep water during the survey period (2008 to 2019) but note the gap in sonar bathymetry data in this area (grey area; Fig.2). A more extensive survey grid next to the glacier revealed the western margin of the lake has a shallower (<5m) shelf extending ~50m from shore (bedrock/moraine composition unknown), whereas the eastern margin is likely to deepen (>5m) steeply within 10m of the shore (note sparser depth points on E margin; see Supplementary S1).

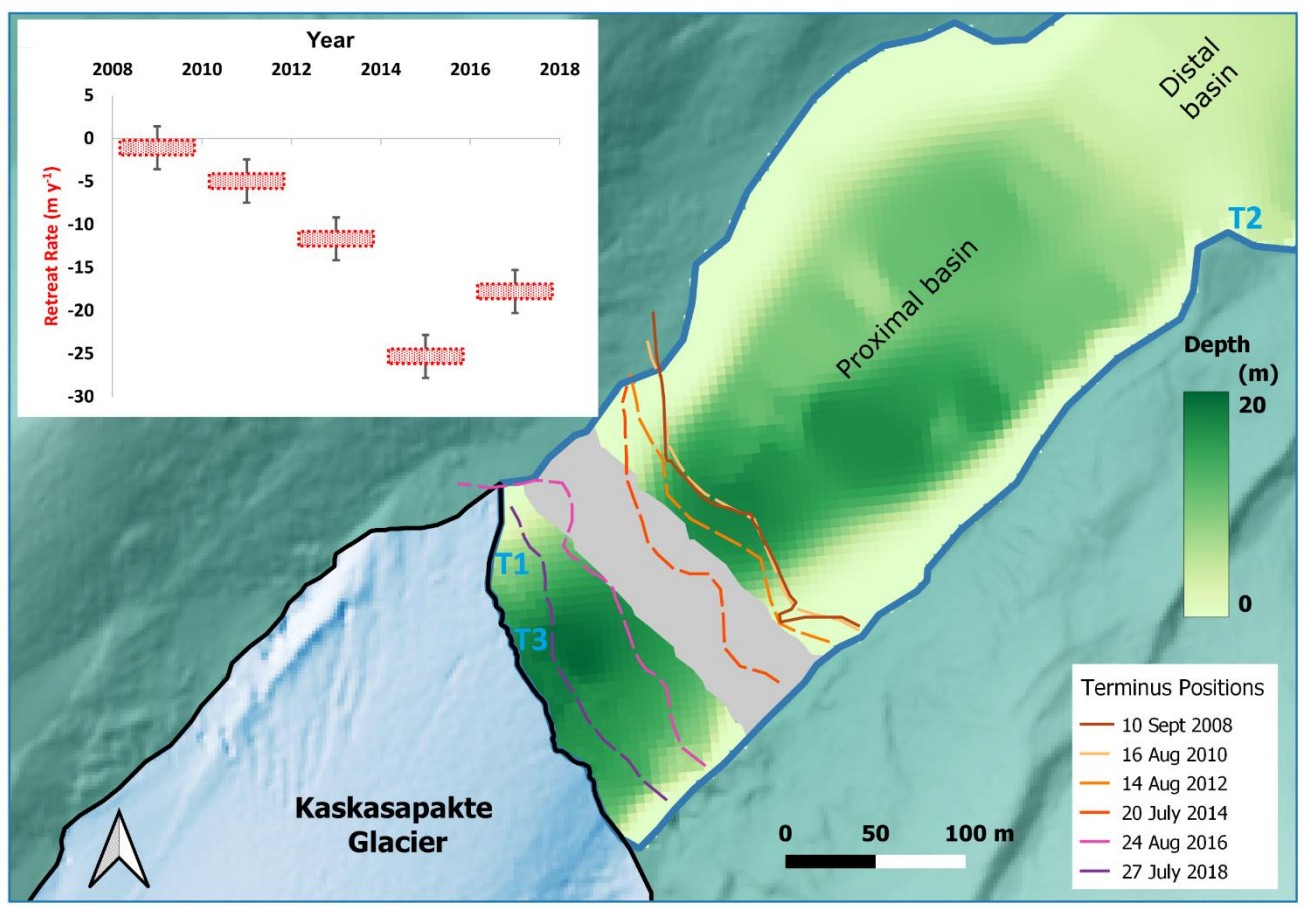

**Figure 2. Digital Elevation model with outline of Kaskasapakte Glacier from 3rd August 2019 and bathymetry from sonar surveys in August 2022. Grey area represents the absence of bathymetry data due to inclement survey conditions. Past terminus positions mapped in QGIS from imagery; i. Lantmateriet (2008) ii. Rapid Eye. Inset shows width averaged retreat rate in metres per year between image acquisition dates. Error bars indicate +/- 1 Rapid Eye pixel (5m). Note T = Thermistor position.**

## 4.2 Subaqueous Terminus Geometry 2019

A strong reflector was returned from the northwest sector of the ice margin during the side-scanning sonar survey on 5th August 2019, which is interpreted to come from the subaqueous ice terminus (Sugiyama et al., 2019). The northernmost sector of the subaqueous terminus produced a strong reflector several metres to the west (up-ice) of the subaerial ice cliff, whereas the central sector showed a strong (but fractured) reflector extending into the lake beyond the top of the subaerial ice cliff as delineated from the DEM (3rd August 2019) (Fig.3). We interpret this as possibly being a small section of subaqueous ice foot protruding beyond the line of the subaerial cliff (by ~5 m) in the central part of the sonar scan (Fig.3a), as has been reported from side scanning sonar of Grey glacier in Patagonia (Sugiyama et al., 2019). Note that confidence in the precise extent (<1 m) of this is low given the positional accuracy of the GLONASS GPS system and associated uncertainty of Garmin Fishfinder sonar systems (+/- 5.9 m at 50 m; Purdie et al., 2016). The southernmost section of the sonar scan is typified by a strong reflector notably recessed underneath (> ~10 m) the subaerial ice cliff (Fig. 3a), corresponding to an undercut cave and englacial outlet (Fig. 3c) from which currents were observed on afternoons (moving ice bergs).

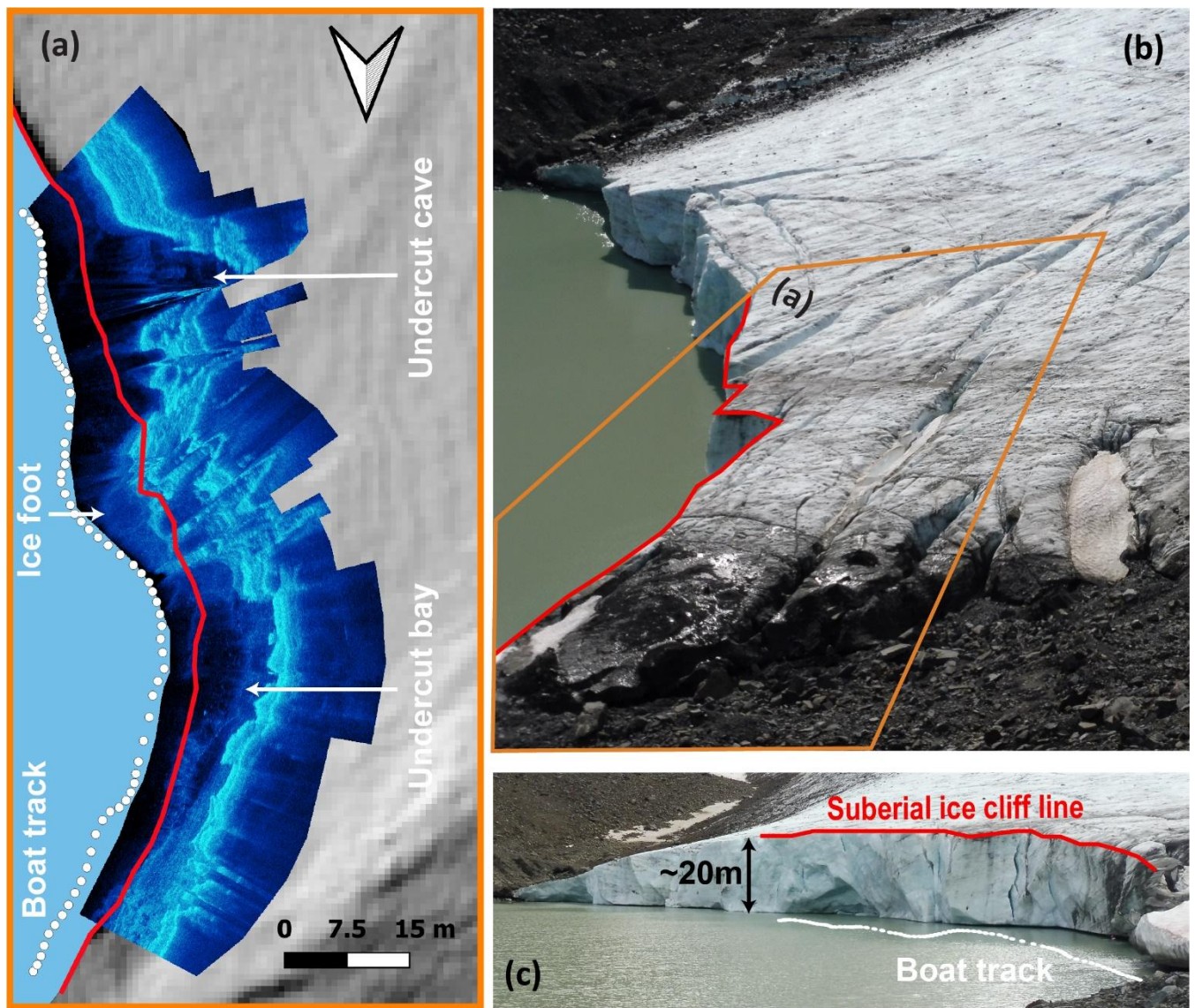

Figure 3. (a) Side scanning sonar (5$^{th}$ August 2019) plotted over SfM DEM (3$^{rd}$ August 2019), with red line denoting the top of the subaerial ice front and white dots denoting survey track of the ASV – Note the cave feature near the southern end of the track line (central part of terminus). (b & c) Images of KG looking south across the terminus (3$^{rd}$ August 2019). Note small crevasse parallel to ice cliff near central portion. Orange inset in (b) denotes area covered by (a).

## 4.3 Terminus Volume Change 2015 to 2019

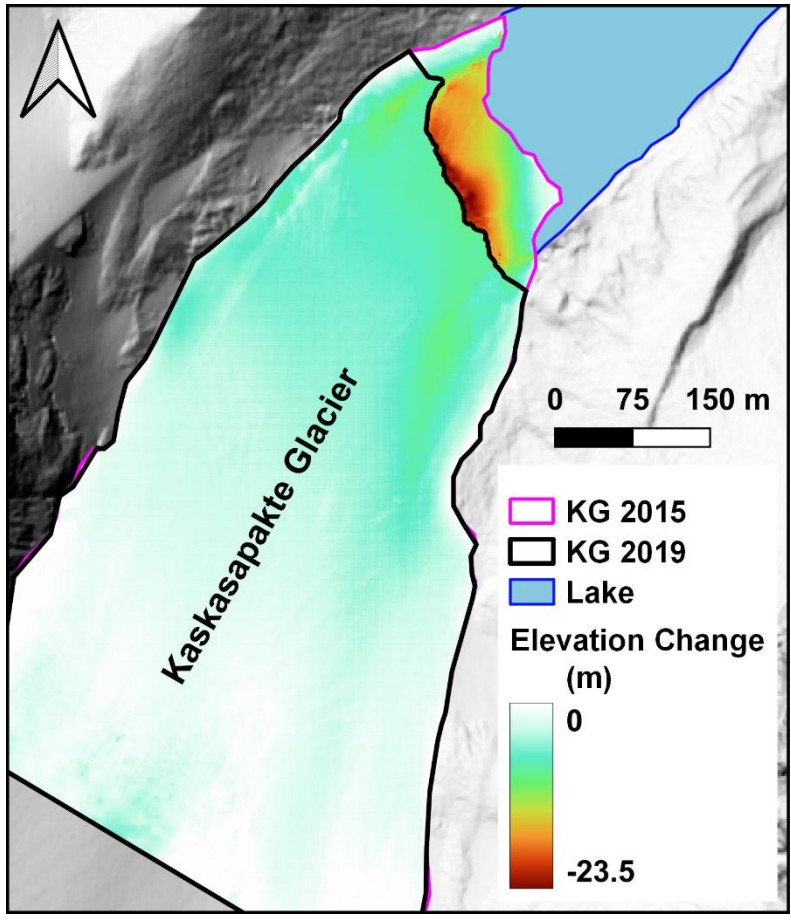

**Figure 4. Surface elevation change between August 2015 (Lantmateriet DEM; c. 1m residual z error) and 3rd August 2019 (SfM DEM; z error = 0.05 m). Black polygon denotes August 2019 terminus area and pink polygon denotes August 2015 terminus (both mapped from contemporaneous DEMs) used in conjunction with bathymetry for volume change analysis in QGIS.**

There was substantial thinning of KG's terminus from 2015-2019, predominantly focused within 200 m (horizontally) of the calving front (Fig. 4), within this area surface lowering tended to be relatively uniform (7 to 8 m). Ice surface velocities derived from NASA ITS_LIVE feature tracking were below $\sim$ 40 m y$^{-1}$ (see S6; Gardner et al., 2019) but high uncertainties prevented incorporation into geometric calculations; so presented volume changes are essentially 'static' as dynamics were not incorporated. Terminus elevation change analysis (section 3.3; RMSE = 0.52 m) was conducted to calculate: (a) volume loss from surface lowering (black polygon) and; (b) volume loss from terminus retreat between 2015 and 2019 (pink polygon). The total surface volume loss (from surface lowering) across the 3rd August 2019 terminus outline (black polygon) since August 2015 was 774,374 m$^3$.

Removal of the 2015 glacier ice surface down to the 2019 lake level, varied between 0 to 23 m across the pink polygon area (Fig. 4). Bathymetry (cf. Sect. 4.2) varied slightly in this section but reached maximum depth of 20 m in the central sections

of where ice had been removed since 2015 (Fig.2). The total volume loss (above and below lake level) between 2015 and 3$^{rd}$
August 2019 (pink polygon) was calculated to be 336,373 m$^3$, thus highlighting the substantial changes in terminus geometry
at KG. The total volume loss (a. and b.) from KG terminus area (across black polygon and pink polygon) between 2015 and
2019 was 1,110,747 m$^3$. Therefore, retreat of the glacier from the lake (cf. Fig 4: volume of ice lost from the pink polygon
area) accounts for 30.2% of surface volume change in the terminus area (black and pink polygon) between 2015 and 2019.

### 4.3.1 Subaerial Geometry Terminus Change 2017

In July 2017 the KG terminus was characterised by substantial (several metres wide) cave features along its south-eastern
section, but a relatively uniform vertical ice-cliff underlain by a thermal notch along the northern section (Fig. 5a). Calving
events captured in time lapse imagery (section 3.3) on 26$^{th}$ July 2017 show icebergs produced from a roof instability in the
main central cave, and more substantial icebergs (several metres wide) from a collapse above the thermal notch (Fig. 5b).
These were exported from the glacier front during afternoons. Following this latter event, no thermal notch was visible at the
waterline along the northern sector of the terminus, however a sharp thermally-eroded notch had reformed along this section
of the terminus by the 4$^{th}$ August 2017. Calving events were observed to be predominantly driven by thermal undercutting
throughout July 2017 (Supplementary Table 1). Crevassing at the ice-front was limited to some relatively minor diagonal
crevasse/weaknesses on the south-eastern sector of the margin (left image margin in Fig. 5 a. and b.).

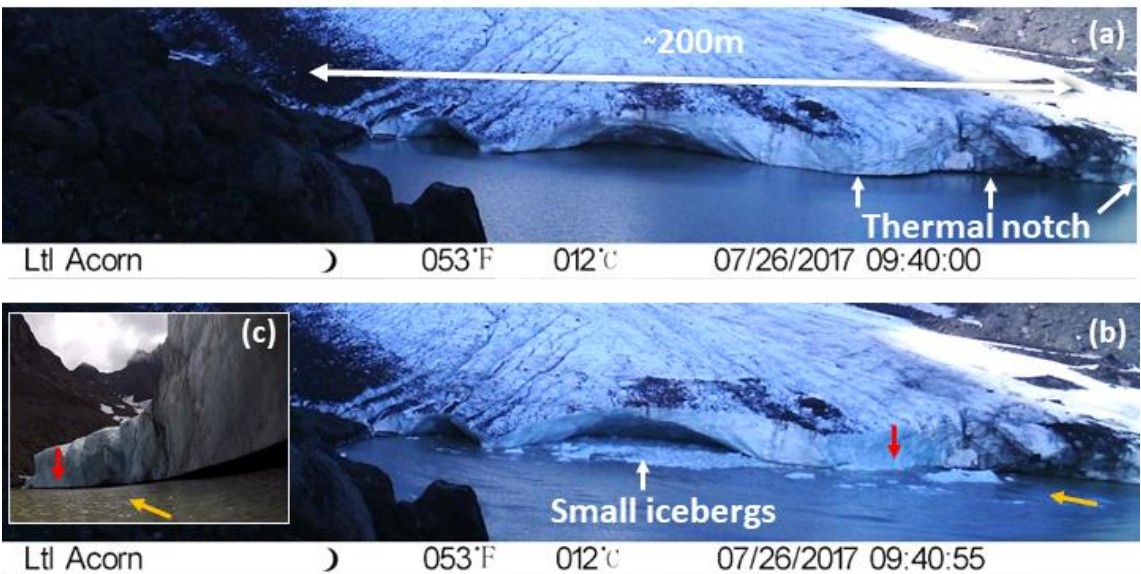

**Figure 5. Timelapse images of calving activity at the glacier terminus in summer 2017: (a) 09:40:00 on the 26$^{th}$ July, showing thermal notch and cave features; (b) 55 seconds later, capturing results of multiple calving events, including collapse of cave roof and detachment of large iceberg above the thermal notch (red arrow). Yellow arrow illustrates orientation of viewpoint presented in panel (c); (c) looking east along the terminus on 4$^{th}$ August, showing shadowing at the waterline indicating lateral extent of thermal**
**notch undercutting the terminus. Note the lack of crevassing and the lateral debris band behind the ice front.**

### 4.3.2 Subaerial Geometry Change 2019

In July-August 2019 the glacier terminus subaerial geometry was characterised by overhanging features and undercuts as the subaerial cliff profile protruded above the waterline, where an extensive thermal erosional notch extended across most of the terminus (note the shadowing in such features in Fig. 6). On 6th August a waterline calving event occurred (by the thermistor
string), which resulted in a cave feature several metres deep (below red star Fig. 6b). Above this cave, cracks developed on the glacier surface, which were observed to widen further during 26th July and eventually failed sometime during 28th and 29th July (Fig. 6c). A relatively small (metres across) cave feature was observed in the central part of the terminus during July/August 2019, which was notably smaller than the relatively large (10's of metres across) central cave feature in 2017. More crevasses were present at the terminus than in 2017, with some orientated diagonally back into the glacier on each margin, as well as
crevassing directly behind and parallel to the ice front (see Supplementary S2).

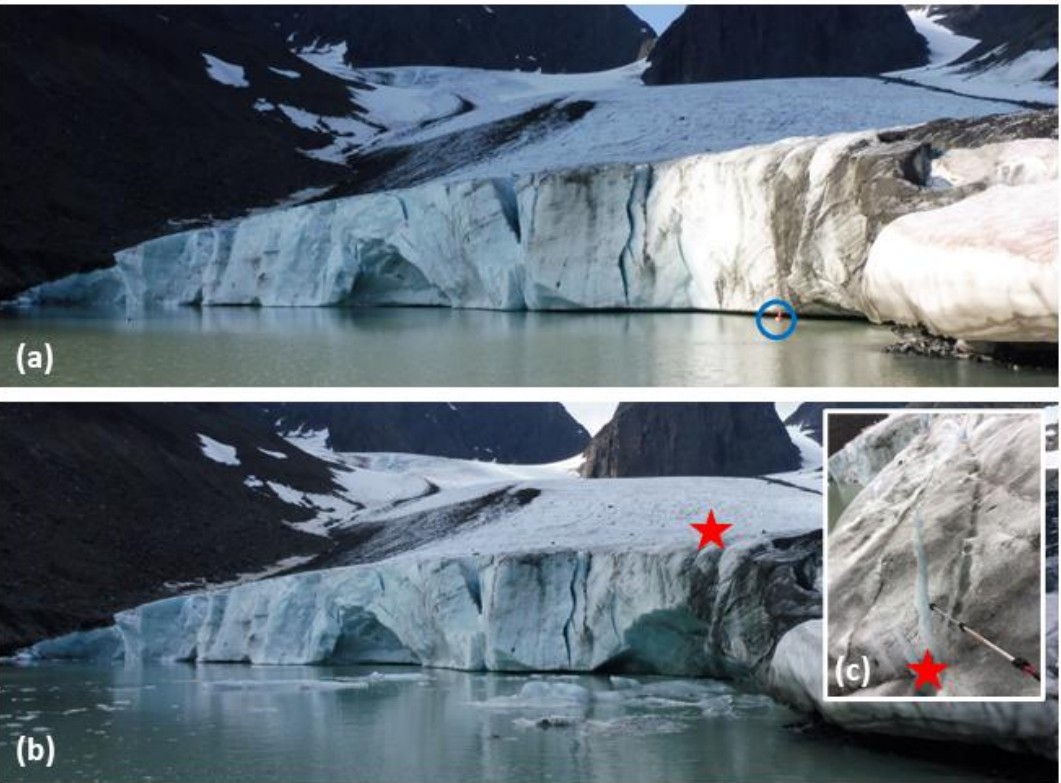

**Fig.6       Images of KG (Panasonic Lumix DMC-TZ57) looking southeast across the terminus with blue circle denoting position of buoy and thermistor string (Fig.7.a) (a) taken at 8:32 on 6th August 2019 (b) 10:03 on 6th August 2019 (c) Development of crack**
**above arch on 6th August 2019. Red star indicates location of cracking in image (c). Note the exposure of pronounced undercutting.**

## 4.4 Calving mechanism classification and environmental drivers from 2019 melt season

Buoyant plumes were not observed at the ice front. During the 2019 melt season calving events were predominantly driven by thermal undercutting creating an unstable ice cliff profile (Fig. 7). During a period of 46 days (5th August to 19th September 2019) there were 41 calving events in total, with 15 (36.6%) sheet collapses, 12 (29.3%) roof/arch collapses, 9 (22%) waterline events, 1 icefall and 4 events that were impossible to define the mechanism. Water temperatures were monitored in the ice proximal area (at 1 m depth; Fig.7) from 29th July to 28th August 2019 (mean = 3.2°C), ice proximal temperatures of 3.6°C were also measured at 20 m depth on 8th August 2019 (Supplementary S3). Lake temperatures of between 3 to 4 °C were also recorded at the central lake point (at 5 m depth; see S5); both observation points suggest the lake was well mixed during the observation period in 2019. Three different phases of calving have been identified through this 30-day time series, with two periods of high calving activity (phases 1 and 3) and a relatively quiescent phase in between (phase 2) (Fig. 7).

Phase 1 from the 5th -16th August was characterised by high-frequencies of calving from a variety of mechanisms, air temperatures of ~3-10°, regular light precipitation events (<10 mm per day), and lake temperatures of ~4°C. Fluctuations in water temperature and light intensity on the 10th and 11th of August correspond with large calving events in the timelapse imagery. Phase 2 from the 16th - 27th August was characterised by very low-frequencies of calving via 1 mechanism (sheet collapse), air temperatures of ~5 - 10°C, mostly regular light precipitation events but with 1 large event (30 mm in 24 hours), and lake temperatures of ~2.5 – 3°C. Phase 3 from the 27th August-19th September was characterised by high frequencies of calving for the first 4 days followed by lower frequencies from a variety of mechanisms. Air temperatures were more varied with highs of 17°C and lows of -4°C, there were irregular light precipitation events (<10 mm per day), with lake temperatures of ~2.5 - 4°C. A calving event on 27th August resulted in a sharp (~0°C) but short (<24 hours) drop in water temperature before water temperatures around the sensor increased to 4 °C. A calving event on 28th August removed the thermistor string, which was likely carried into the lake body but was not directly warmed by solar radiation until 7th September (Lux remained <2,500).

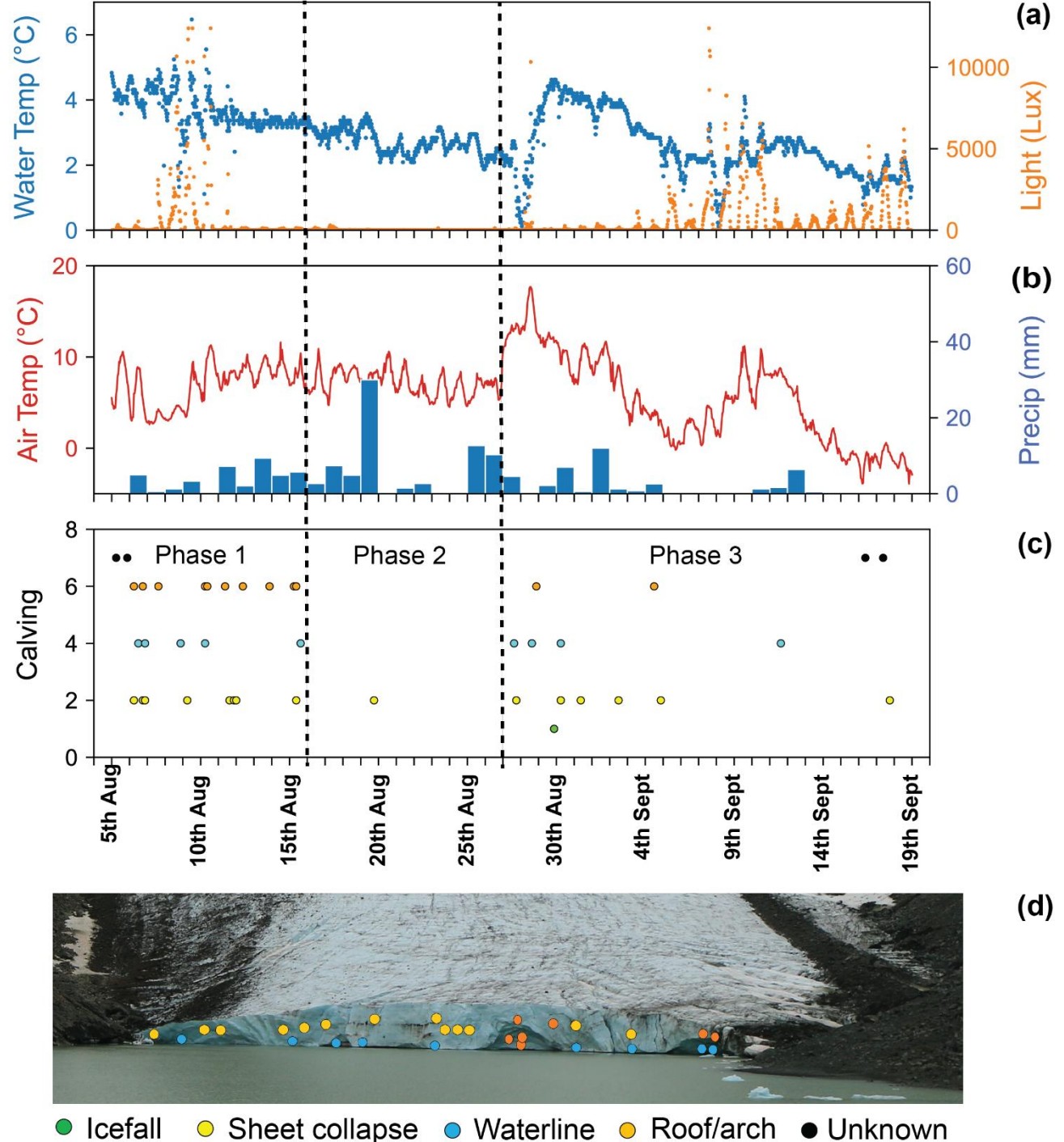

Figure 7. Lake, climate and calving parameters for 5th August–19th Sept 2019. (a) Water temperature data from  KGL from thermistor (+/- 0.5 ºC) at ~1 m depth next to the glacier terminus, with light intensity (Lux). (b) Air Temperature from SMHI AWS at Tarfala Research Station Lower. Precipitation data from AWS at Tarfala Research Station (c) Classification of calving events

(following classification methods of How et al., 2019) (vertical axis is class of calving). Points are plotted by calving class and time (3-hour window) of calving activity. Stippled lines plotted to identify periods of calving activity or quiescence. (d) Location of calving events on the terminus over the duration of the survey.

## 5. Discussion

### 5.1 Terminus retreat

KG has undergone the most pronounced terminus retreat (126 m) of any glacier in the Kebnekaise area between 2010 and 2018 (Dye et al., 2022). NASA ITS_LIVE feature tracking suggests ice surface velocities (~100m behind KG terminus) remained relatively low (< 40 m y$^{-1}$) during this period (see S6) (Gardner et al., 2019), so variations in velocity are unlikely to have substantially affected the retreat of KG between 2015 to 2022. The retreat of the land terminating Isfallsglaciären (122 m) was also large over this period, however, this was mainly due to enhanced mass loss from the decay of an icefall covering a relatively large proportion of the terminus area (Table 2). In contrast Rabot's glacier (also land terminating) only retreated 81 m between 2010 and 2018 (Dye et al., 2022). Whilst the relatively muted retreat of KG between 2008 and 2012 (Fig.2) cannot be fully explored here, we argue that subsequent warm events and heatwaves during the summer (particularly 2014 and 2018) have enhanced the terminus retreat through increased subaqueous melt and thermal undercutting of the terminus (Dye et al., 2020). Indeed, the width averaged terminus retreat of six neighbouring land-terminating glaciers around the Tarfala valley from 2015 to 2022, was only 7.3 m y$^{-1}$, which is roughly a third of the retreat rate at KG between 2014-2016 (25.31 m y$^{-1}$) and 2016 to 2018 (17.77 m y$^{-1}$) (Houssais, 2023). We argue that contact with KGL has enhanced the retreat of KG substantially and the processes (and balances) affecting this recession are discussed further below.

**Table 2;** The 6 largest width averaged retreats (following Lea et al., 2014) (2010 to 2018) of glaciers around the Kebnekaise area, mapped from Rapid Eye imagery (Dye et al, 2022). * denotes glaciers in contact with a proglacial lake.

| Glacier | Width Averaged Retreat | Retreat rate 2010 to 2018 |
|---|---|---|
| Kaskasapakte * | 126 m | 15.75 m y$^{-1}$ |
| Isfallsglaciaren | 122 m | 15.25 m y$^{-1}$ |
| Mårmaglaciären * | 110 m | 13.75 m y$^{-1}$ |
| Rabot's glaciär | 81 m | 10.13 m y$^{-1}$ |
| Östra Bossosglaciaren | 67 m | 8.38 m y$^{-1}$ |
| Riehppiglaciaren | 63 m | 7.88 m y$^{-1}$ |

The terminus geometry of lake terminating glaciers are controlled by glacier velocity, subaerial melt rates, lateral constraints/support from bedrock/moraines, subglacial bedrock geometry, mechanical processes and thermo-erosional processes (Carrivick and Tweed, 2013). Lake depth and surrounding bedrock geometry can determine how much support and back stress a glacier receives from topography and so are an important control on the calving rate ($R^2 = 0.83$ for calving rates to water depth of 9 glaciers; Warren and Kirkbride, 2003; Boyce et al., 2007). Sonar surveys suggest the bathymetry of the ice

proximal basin (lake bed) is fairly consistent, with shallower sections along (<5 m) the margins and deeper central areas of 15 to 20 m (Fig. 2). Unfortunately, a gap (~50 m across) in the sonar survey of KGL remains, but we suggest that the gap

bathymetry is likely to be similar, as it is too small to have a substantial over-deepening and there is no obvious evidence for any substantial shallowing (such as a bedrock riegel or moraine) as large icebergs passed to the lake mid-point. We argue that the main back stress acting on the terminus is likely to be via lateral support from the surrounding moraine, although this is largely ice cored and unstable material. Lake temperatures (cf. section 5.3) and associated calving mechanisms (cf. section 5.4) are more likely to be the main drivers of recent retreat rates at KG rather than variation in lake depth or lateral (frictional)

support. Although this may change as the glacier retreats further out of the lake basin (Sutherland et al., 2020).

## 5.2 Surface Volume Change 2015 to 2019

There has also been substantial surface lowering of KG between 2015 and 2018, with changes of 7 to 8 m extending 200 m back up glacier across the front, with 774,374 $m^3$ reduction in glacier volume in the 2019 terminus area between 2015 and 2019 (RMSE = 0.52 m; Fig. 4). Whilst a large proportion of this is likely to be due to subaerial melt, the contribution of

330 dynamic thinning from glacier velocity has not been constrained. We report ice surface velocities < ~40 m $y^{-1}$ from NASA ITS_LIVE (Gardner et al., 2019), which are similar to glacier velocity observations from nearby Storglaciaren (10 to 30 m $y^{-1}$) and Rabot's glaciär (6 to 12 m $y^{-1}$) that are relatively low due to frozen margins on both glaciers (Brugger et al., 2007). We argue observations of water filled crevasses persisting at KG margins through the melt season suggests the margins are frozen to the bed (Supplementary S2) (Moore et al., 2011). The volume of ice lost into the lake (336,374 $m^3$) between 2015 and 2019

is a conservative estimate, given that ice flux to the front is not constrained (due to lack of high resolution velocity data). Yet this represents 30.2 % of the overall geometric volume change at KG terminus between 2015 and 2019. So a greater understanding is needed of how lake-glacier interactions affect mass balance processes, and glacier responses to future climatic changes.

## 5.3 Proglacial Lake Temperatures

Previous melt models for lacustrine terminating glaciers have been compromised by a lack of data from the hazardous water to ice contact point, and  past studies have reported relatively low uniform temperatures (e.g. 1 °C, Truffer and Motyka., 2016). We report temperatures of 4 °C directly at the ice-water contact point (over 11 days) following numerous iceberg calving events above thermal notches during summer 2019 (Fig.5). Contemporaneous water temperatures of 3.5 °C were also recorded at 20 m depth next to ice front and at 5 m depth at the central lake point (see supplementary), which we argue shows the lake

to be relatively well mixed during the period of field observations in 2019. Near surface (1 m) lake water temperatures were significantly correlated ($R^2$= 0.47) to air temperature (measured at Tarfala Research Station ca. 5 km away) during August/September 2019 (Supplementary S4), which is similar to studies of other Arctic lakes (e.g., $R^2$ = 0.52; MacIntyre et al., 2009). A close relationship between seasonal lake water temperature and seasonal ice front position has also been reported for Glaciar Perito Moreno in Patagonia ($R^2$ = 0.96) (Minowa et al., 2017). We argue that elevated proglacial lake water

temperatures (and circulation) resulted in rapid thermal notch development and a series of iceberg calving events at KG in 2017 and 2019 (Röhl, 2006). There is a growing body of evidence that the thermal structure of proglacial lakes can have a substantial influence on glacier retreat rates (Sugiyama et al., 2021). The relatively high proglacial lake surface temperatures observed in this study would only partly account for rapid thermal undercutting of the notch, and it is proposed that currents flowing along the ice-water contact point would account for the remaining heat transfer required to create such a melt feature in a short space of time (9 days) in 2017 (Röhl et al., 2006).

## 5.4 Calving Mechanisms

The response of Arctic proglacial lake thermal regime to warmer air temperatures has received relatively little attention to date (at the process scale), despite rapid warming in the region (and heatwaves) that will likely warm proglacial lakes and consequently increase subaqueous melt and terminus undercutting (Kim et al., 2018; Rantanen et al., 2022). Examining the timing of changes in calving terminus geometry is critical in order to understand the interlinkages within the system. Lag times must also be considered, as the thermal undercutting process takes place over days (and longer); with Röhl (2006) reporting a maximum thermal notch development rate of 65 cm day$^{-1}$. So once calving activity above thermo-erosional undercutting has reformed the ice front to a more stable vertical profile, there is likely to be a quiescent phase (with minimal calving) until further thermo-erosional undercutting produces a vertical profile unstable enough for calving to ensue. We argue that the calving regime at KG has been dominated by thermo-erosional undercutting during the observation periods; where periods of high calving activity have created a more stable vertical ice cliff profile and lead to a quiescent phase of low calving activity. Subsequent changes in limnological conditions at the ice front (both in 2017 and 2019) then resulted in further substantial (metres) thermo-erosional undercutting of the terminus followed by a period of high calving activity.

### 5.4.1 2017 Field Season Changes in Glacier Front

There was strong evidence for thermo-erosional undercutting dominating terminus geometry changes during the 2017 field season, including overhanging caves and a thermal notch across the whole terminus (Fig. 5). We argue that the export of icebergs from these caves during afternoons (captured in timelapse) suggests some englacial drainage through them. The sharpness of the notch (Fig. 5c) suggests formation through subaqueous melt (subaerial melt leads to rounding of features). This unsupported terminus geometry resulted in multiple calving events above the original thermal notch, producing a vertical terminus subaerial profile and no thermal notch at the waterline (Fig. 5b). Following this initial calving, rapid thermal notch formation was observed over a 9-day period between (26$^{th}$ July to 4$^{th}$ August), demonstrating that terminus geometry can be quickly undermined to produce an unstable ice cliff profile (Fig.5c). These observations of calving above thermally undercut features during July and August 2017 coincided with the second highest terminus retreat rate (17.77 m y$^{-1}$; 2016 to 2018). We argue that thermal erosional undercutting was the main process driving mass loss and terminus retreat during this period in 2017, particularly as the lack of crevassing behind the ice front (Fig.5) and relatively low velocities ($< \sim 40$ m y$^{-1}$) suggests that

there was no contribution of mechanically derived processes on the fracture pattern and consequent production of icebergs at this time.

### 5.4.2 2019 Field Season Changes in Glacier Front and Environmental Conditions

During the 2019 field season we categorised 3 phases of calving activity (How et al., 2019); with high numbers of calving events during phase 1, a more quiescent phase 2, followed by a period of moderate calving activity in phase 3 (Fig. 7). Calving events on the 8th, 9th and 10th of August caused short term (<1 day) distinct cooling (>1 °C) of water at the terminus, suggesting such cooling tends to be short and sharp (if icebergs are removed quickly from the terminus). Water temperatures remained relatively high (fluctuating ~4 °C) during phase 1, before a sharp decrease to 3 °C shortly (~24 hours) before the beginning of the quiescent calving phase 2 (Fig.7). Air temperatures were relatively similar over this period, with the latter fluctuating between 3 and 10 °C during phase 1 and between 5 to 10 °C during phase 2 (Fig.7). Therefore, we argue that the drop in water temperatures at the end of phase 1 is likely due to input from the glacial drainage system (with some internal lag likely), following melt and precipitation in the days prior to 15th August, causing cooling and mixing of lake water. Lake water further cooled following a large rainfall event on 19th August 2019. The timing of decreases in water temperature at the end of phase 1 (16th August) and cessation of calving roughly 24 hours after, would tentatively suggest that calving is partly controlled by lake water temperature but other factors need to be considered.

During phase 3 water temperatures increased to 4 °C, which we argue is due to increased air temperatures (maximum of 17.3 °C on 28th August) combined with a period of relatively high winds (>5 m s$^{-1}$) that resulted in efficient positive sensible heat flux into the lake (see Supplementary S5). This is an important increase in lake temperatures relatively late in the melt season as it corresponded with a period of iceberg calving activity that we argue was due to meteorological conditions (rather than solar insolation) warming the lake and driving further undercutting of the terminus.

Side scanning sonar confirmed that undercutting (by several metres) was relatively extensive across the terminus (Fig.4), particularly along the northern section of terminus where subsequent calving activity occurred above an undercut several metres deep (Fig.6). Analysis of calving mechanisms in 2019 suggests that calving was primarily driven by thermal undercutting; 36.6% of calving events were sheet collapses, likely to be caused by weakness at/near the waterline, and 22% were waterline events, often occurring above or below a notch (How et al., 2019). There was also a large proportion (29.3%) of calving events from cave roof/arch collapses, which are caused by undercutting at the ice front that develops overhanging features where cliff support has been removed (Fig.6). Note that icebergs were also observed to be exported from these cave features during afternoons, suggesting some englacial drainage through them. There was some clustering of calving activity in 2019, with some events happening just hours apart and these tended to be a waterline event followed by a sheet or arch collapse higher up the ice front (Fig.7). This analysis provides support to calving being primarily driven by thermal undercutting of the ice front during the observation period. Whilst variability in mechanical support due to lake depth cannot be conclusively eliminated (due to data gap in the bathymetry) we argue that mechanical calving processes are not the main driver of recession of KG terminus, due to the factors discussed above and factors further discussed in the next section.

### 5.4.3 Mechanical calving processes

The depth of the lake bed can be highly important as thinning of the glacier snout makes the terminus more susceptible to mechanical calving processes. As the ice mass thins it becomes closer to the flotation point, which can increase crevassing due to upward forces (Benn et al., 2007; Boyce et al., 2007; Tsutaki et al., 2013). Any upward forcing on the glacier could enable lake water penetration under the glacier and if subglacial water pressures increase it could increase glacier velocity and crevassing from extensional flow (Sugiyama et al., 2016). Crucially, these mechanical processes are unlikely to be the primary drivers at KG between 2017 and 2019, as evidenced from the minimal crevassing behind the ice front and consistent relatively low ice surface velocities ($< \sim 40$ m y$^{-1}$) (Fig. 3 and 5). There were some prominent crevasses at the terminus during 2019 and we argue that these were not the primary driver of iceberg calving as demonstrated by the analysis of calving mechanisms (Fig. 7), although they sometimes controlled the extent and size of icebergs. We also argue that some of these crevasses that developed behind and parallel to the ice front were localised and have developed in response to undercutting of the terminus and consequent stresses on ice behind the calving front (Iken, 1977) (Fig. 3).

Unfortunately there is no existing radar survey to confirm the thermal structure of KG. However, we argue that the margins and terminus are likely to be frozen to the bed (as at neighbouring Storglaciaren) as water filled crevasses persisted through the melt season and the supraglacial debris bandsseen near the terminus of KG (see Supplementary 6) are typically associated with the transition of temperate to cold based ice (Moore et al., 2011; Monz et al., 2022; Carrivick et al., 2023). Furthermore, ice surface velocities remained relatively low ($< \sim 40$ m y$^{-1}$) during the period (S7) (Gardner et al., 2019). Lake water penetration and hydraulic conductivity to subglacial drainage system is likely to be limited, so mechanical drivers of calving are likely to be minimal, at least initially. Whether this cold based ice has been removed from the front through iceberg calving is an open research question that is of critical importance, as glacier velocity will increase if the resistance from cold based ice at the terminus is removed and may occur at other Arctic polythermal water terminating glaciers. Therefore increased thermal undercutting of lacustrine glacier termini due to proglacial lake warming needs to be better understood given increasing air temperatures, not only for the initial enhanced volume loss of such glaciers (as argued in this study) but also potential implications for glacier dynamics.

### 5.5 Wider Implications

Whilst the observations from KG during 2017 and 2019 are a relatively limited snapshot in time; they do show that water temperatures were warmer ($>4$ °C) than has previously been expected (1 °C) for small proglacial lakes and resulted in substantial (metres) thermal undercutting with periods of high calving activity in both melt seasons. This rapid undercutting has been focused around englacial conduits observed in the glacier terminus, that create crucial weaknesses in the ice front and also a focus point for driving water circulation during periods of high meltwater output. This study provides an observational overview of the system and future research should focus on the evolution of subaerial and subaqueous lacustrine glacier termini

(in conjunction with lake thermal stratification), particularly given future predictions of air temperature increases and associated reduction of lake ice cover duration, which will expose lake water to greater warming influences (Huang et al., 2022). Annual lake temperature records would constrain lake thermal regime, which if combined with repeat high resolution digital surface models (subaqueous and subaerial) would allow greater confidence in understanding proglacial lake forcing of glacier changes; particularly if combined with high resolution velocity data. Side scanning sonar surveys of the subaqueous ice front has considerable potential to constrain subaqueous melt rates of lacustrine termini and subsequent evolution of geometry in relation to calving activity (from timelapse imagery). Glacier terminus geometry and lake conditions during quiescent phases of calving should also be carefully observed in order to understand what changes in balances at the ice front may trigger periods of frequent calving activity. Comprehensive sonar surveys of the ice proximal lake bed are essential to identify any possible shallower areas that may have provided sufficient support to maintain terminus position stability. This combined with monitoring in situ meteorological parameters would provide a strong basis for understanding the feedbacks between climatic changes and processes of mass loss at lacustrine glacier termini; which is currently poorly constrained in glacier mass loss models (Carrivick et al., 2020). This is essential in not only predicting the future mass balance of glaciers in contact with proglacial lakes, but also lake thermal regime and subsequent impact that any changes have on their ability as a carbon sink, as well as impact on stream hydrology and ecology (Carrivick et al., 2020, Fellman et al., 2014, St Pierre et al., 2019).

## 5.   Conclusion

The water temperatures reported by this study are substantially warmer than the uniform 1 ºC that has been previously assumed for small proglacial lakes (Truffer and Motyka, 2016). This study provides the first (to our knowledge) direct field evidence of an Arctic glacier retreat being enhanced by contact with warm (>4 ºC) proglacial lake temperatures, through rapid thermo-erosional undercutting and associated calving. Further adding to observations of warm (>4 C) proglacial lakes in other regions of the world (Kirkbride and Warren 2003; Röhl, 2006; Sugiyama et al., 2016; Watson et al., 2020). KG lost 1.5% of the surface area between 2008 and 2018, with retreat rates increasing after 2012 and reaching a maximum of 25 m per year. Processes of ice loss at the proglacial lake to glacier interface were a key part of this terminus retreat as 30% of ice volume loss between 2015 to 2019 was from frontal ablation as the terminus position retreated across the proglacial lake. Processes arising from contact with the proglacial lake have had a substantial role in volume loss and retreat of KG.

Thermal undercutting was found to be the primary driver of iceberg calving during 2017 and 2019. The rapid (~9 days) formation of thermally eroded notches at the waterline in 2017 highlights the short time frame in which the subaerial ice front can be undermined. Calving events mainly occurred during the early summer in 2019, with a quiescent phase before more calving activity occurred in late summer, suggesting several weeks are required for thermal undercutting to become extensive enough to trigger calving at this glacier (depending on meteorological conditions). Thermal erosional undercutting of the terminus appeared to be extensive during 2019 and was also confirmed to be several metres deep from side scanning sonar

imagery. The general lack of a subaqueous ice foot and extensive undercutting resulted in subaerial calving above this and some crevassing parallel to the ice front, in response to changes in the stress regime (Iken, 1977). We argue that the influence of mechanical processes on calving (such as topples and ice fall calving from crevassed areas) have been minimal during the period of observations, whereas thermal undercutting from warm proglacial lake temperatures has been extensive. During this period there has been a series of heatwave events, with July 2018 representing the warmest (5.6 ℃ above the long-term average) and also the magnitude of the event has been strongly attributed to anthropogenic forcing (Jonsell et al., 2011; Yiou et al., 2020). Given the observed and projected air temperature increases in the Arctic, it is of critical importance to further study the distribution and impact of proglacial lakes on glacier retreat across the region (Carr et al., 2013).

**Author Contributions**

Conceptualisation, A.D., R.B., F.F., J.M. and D.R.; Methodology, A.D., R.B., F.F., J.M., D.R, and M.B.; Investigation, A.D., R.B., F.F., J.M., M.D. and M.B.; Resources, A.D., R.B., F.F., J.M. and D.R, Data curation; A.D., J.M., N.K. and M.B,, Writing – original draft preparation, A.D., Writing – reviewing and editing; A.D., R.B., F.F., J.M. and N.K.; supervision D.R. and R.B.; project administration, University of York; funding acquisition, D.R. and A.D. All authors have read and agreed to the published version of the manuscript.

**Competing Interests**

The contact author has declared that none of the authors has any competing interests.

**Acknowledgements**

This research would not have been possible without the excellent support and logistics from Tarfala Research Station and funding from INTERACT Transnational Access (CIWB 2019 and GLRETA 2020) and the Royal Geographical Society (SRG 2020). We would also like to thank Martyn Dye, Steve Thompson and Tom Sloan for enduring several days of poor weather whilst at the field camp in Sweden and still providing excellent field support.

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
