# Peer review of "Warm proglacial lake temperatures and thermal undercutting enhances rapid retreat of an Arctic glacier"

_EGUsphere, 2024_

## Author Comment (AC1)

**Response to Review 2; Kas' glaciar paper**

**General comments.** The effects of ice-marginal lakes are a topical and emerging area of research. Due to the scarcity of glacial lake observations, the authors make a strong case for investigating physical lake properties and their impacts on ice mass loss. This is a nicely designed and detailed study which reports a holistic analysis of lake temperature, calving events, lake bathymetry and calving front geometry from an ice-marginal lake in Arctic Sweden from several time periods within the last decade. This work provides an interesting and valuable dataset but perhaps the novelty could be teased out a little more (see specific comments). Overall, the manuscript is well-written. I suggest mostly minor corrections which I expect can be addressed very easily. These suggestions are merely intended to help tighten up the precision of the text, but otherwise I support it's publication.

**Authors response;** Thank you for the detailed and carefully considered review of the manuscript. The synthesis and composition of several field seasons at an unstudied remote field site has been somewhat challenging and all comments are greatly appreciated. We will carefully revise the manuscript in light of these and greatly appreciate the suggested improvements.

**Specific comments.** I understand the value of the study, so I focus my review on the data and the methods. A small concern is how lake temperature is reported. The authors state that temperatures of >4 °C are 'warmer than expected'. But what constitutes as 'warm'? Where has this assumption come from, given that given that the same authors in Dye et al. (2021) also find 'warm' lake temperatures (albeit from satellite analysis), and similar temperatures have been reported elsewhere (as is stated in the introduction (line 33). Perhaps the authors could frame their objectives more explicitly to test a hypothesis between what they 'expect' from their own previous analyses vs uniform temperatures of 1 °C reported in the literature?

**Authors response;** We greatly appreciate the support of the value of the study. The uniform 1°C assumption comes from Truffer and Motyka (2016) – we shall review this statement in the Introduction and put it more clearly in context with other studies;

"Whilst previous studies have assumed small proglacial lakes to be a uniform 1°C (Truffer and Motyka, 2016), proglacial lake temperatures of >4 °C have been reported in Nepal, New Zealand, Patagonia and Arctic Sweden (Kirkbride and Warren 2003; Sugiyama et al., 2016; Watson et al., 2020; Dye et al., 2021)."

I think it needs to be acknowledged more clearly that these 'warm' temperatures have been measured at just one single point in the lake and over a short time period (~6 weeks). Given this limitation, it is a shame that stratification/mixing can't be considered. It seems a relatively shallow lake, so some inferences could be made between both the *depths and* temperatures of other lakes that have been recorded before making direct comparisons.

**Authors response;** A very prescient point. We chose to just present data from the ice front for 2019 for simplicity, as we felt this highly likely to be the coldest point across the lake at the time and also the most pertinent with regard to ice melt. We have also presented temperature readings of 3.5°C from ~20m depth 8th August 2019 in the supplementary (S3) – so we believe the lake to have been well mixed at this time.

We also have in situ lake temperature data for 2017 and 2022 – but felt this would potentially overcomplicate the manuscript (possibly best published in a more limnology focused paper?). Summaries of this are;

- 2017 – temperature gradient across lake in July (3 C at front, 8 C at distal end) – rainfall event mixed the lake to uniform 3.5 C.
- 2022 – lake had already mixed to uniform 3 C by 10th July. Max temperature reading ~4 C.

Some other thoughts are that the stage of lake evolution is important and could be commented on. When did the lake form? As the glacier begins to retreat out of its basin it could be that lake temperature has less of an effect on calving and ice dynamics.

**Authors response;** Yes another very pertinent point (I have a lot of thoughts on this but need several glaciers to demonstrate it on). We shall add/expand comments to '2. Study Area' section line 67 – with reference to the mapping from 1959 in Dye et al. (2022).

***Technical corrections.***

Line 19. 'Scandinavian proglacial lake'/ 'Arctic Sweden'. Ensure consistency with terminology throughout if you can.

**Authors comments;** Thanks we shall review this.

L21-24. Last sentence of abstract reads more as rationale rather as wider implications, I would make it more specific to the study or move higher up in the abstract.

**Authors comments;** Excellent point – we shall move this to the second sentence of the abstract.

L30. Can these citations be placed next to each location rather than grouped at the end of the sentence?

**Authors comments;** Yes I think this would be more useful.

L31. There are 6 instances of Roehl et al. (2006), I think it should be Röhl (2006).

**Authors comments;** Thanks we shall review this.

L35. Could you explain more explicitly how thermal notches promote ice berg calving

**Authors comments;** Thanks we shall amend this to;

" High subaqueous melt rates remove mass from the glacier terminus and cause thermal undercutting (producing thermal notches)  that promotes iceberg calving *through failure of overhanging subaerial ice cliffs* (Iken, 1977; Warren and Kirkbride, 2003; Roehl et al., 2006).

L38 – 'Such high subaqueous melt rates remove ice from the terminus' –  this is repeated from line 34 so I would move this explanation higher up and rephrase

**Authors comments;** Thanks we shall amend this to;

"Proglacial lake temperatures have been found to control seasonal ice front position in Patagonia and terminus morphology in New Zealand, where rapid (0.65 cm d$^{-1}$) thermal notch development has led to substantial undercutting of glacier termini (Minowa et al., 2017; Roehl et al, 2006). *Such thermal undercutting* alters the stress balance at the ice front, as support is removed from the subaerial ice front so stresses increase within the ice, which may either fracture above the overhang (calving icebergs) or develop crevasses parallel to the ice front (Iken, 1977)."

L45. Do you mean downstream temperatures?

**Authors comments;** Yes

L48. I would argue there has been quite a bit of attention recently

**Authors comments;** We feel that in proportion to the size and number of glacial lakes across the Arctic there has been relatively little attention focusing on case study scale glacial-lake interlinkages – Mallalieu's work in Greenland, Carrivick's work in W Greenland, some small studies in Norway. Beyond that are mostly regional based (mostly remote sensing?) studies in Greenland and Novaya Zemlya. I've had a quick search, but couldn't find anything further but may have missed something – please let us know if we have missed some studies!

We could amend line 48 to;

'To date *there have been few processed based field studies at* proglacial lakes *in the Arctic*, despite being associated with enhanced glacier retreat rates around the Greenland Ice sheet and Novaya Zemlya (MacIntyre et al., 2009; Carr et al., 2017; Carrivick and Quincey 2014; Carrivick et al., 2017; Mallalieu et al., 2021; Carrivick et al., 2022).'

L65. ~2,000 m – add a.s.l.

**Authors comments;** Thanks we shall amend this.

L68. ~1916 – add CE

**Authors comments;** Thanks we shall amend this.

L70. Could you add a few more specifics about the study site, e.g. does the lake free over in winter? Is there a lake outlet?

**Authors comments;** Thanks we shall amend line 66;

"KGL is situated at 1,100 m a.s.l. and was 670 m long, with a surface area of 0.13 km$^2$ in August 2014 *with an outlet at the northernmost point and freezes over during winter* (Dye et al., 2022)."

L102. Was any post-processing software used for the sonar data, and did you post-process this data yourselves?

**Authors comments;** Thanks for identifying this oversight, we shall add to line 102;

*'The side view sonar images were viewed and trimmed in SonarTRX software before being exported to QGIS.'*

L146. I think this is the first time an englacial conduit is mentioned so it came a little surprised. Perhaps you could introduce it in the study site section?

**Authors comments;** Thanks we shall amend this and add the following sentence at the end of the study area section;

*'At the start of fieldwork in 2017 the glacier termini was observed to have two englacial conduits at lake level, that became active most afternoons as demonstrated by export of icebergs.'*

L151. A more extensive survey grid next to the glacier revealed the shallow (margins of the lake to be relatively limited, extending out <10 m from the eastern shore… - what do you mean by this?

**Authors comments;** Thanks for the clarification we shall amend this to;

'A more extensive survey grid next to the glacier revealed *the western margin of the lake has a shallower (<5m) shelf extending ~50m from shore, whereas the eastern margin deepens (>5m) steeply within 10m of the shore (Supplementary S1).'*

L170. Is 'cave' a commonly used term to describe ice geometry in this way?

**Authors comments;** Interesting point. We weren't sure of the best terminology here, but 'cave' seemed fitting where; depth>height (for large undercut features). Where depth<height we refer to arches. We are open to suggestions on this and will revisit How et al. (2019). We shall also consider defining these features in the Methodology.

L187. 'Removal of the glacier ice surface (down to lake level) was between 0 to 23 m from removal of ice from the 2015 terminus extent (Fig. 4a).' – I'm not sure I follow this? Would suggest rephrasing.

**Authors comments;** Apologies for that. This section has been very tricky to get right and we will delete the sentence on line 187 and amend it some more to;

*"The area of ice removed from the terminus since 2015 is represented by the pink polygon (Fig.4), some of this ice was relatively close to the lake water level (0 m; Fig.4) whilst sections near the central part of the 2019 terminus were ~23m above lake level."*

L217. 'metres across' - I would rephrase to 'meters wide' instead

**Authors comments;** Thanks. Shall do.

L377. You could also add the following citation in here: Carrivick, J. L., Smith, M. W., Sutherland, J. L., & Grimes, M. (2023). Cooling glaciers in a warming climate since the Little Ice Age at Qaanaaq, northwest Kalaallit Nunaat (Greenland). *Earth Surface Processes and Landforms*, *48*(13), 2446-2462.

**Authors comments;** Thanks! I hadn't seen that paper, which looks to be a very interesting and pertinent read! Reference added.

L392. I'm not sure how you can constrain melt rates from side scan sonar data?

**Authors comments;** Excellent point. One that I am hopefully going to continue to develop – I would rather not expand on it here though if possible? We can remove the statement if needed.

L402. I would reiterate again that several studies of 'warm' lake temperatures have been reported elsewhere and cite these studies (Himalaya, New Zealand etc).

**Authors comments;** We agree and will add the following sentence;

*"Further adding to studies documenting warm (>4 C) proglacial lakes in other regions of the world (Kirkbride and Warren 2003; Röhl, 2006; Sugiyama et al., 2016; Watson et al., 2020)."*

**Figures.**

Figure 1. an inset map to panel (a) might be useful

**Authors comments;** Thank you for the suggestion, but we feel the extent of inset (a) and the grid lines are sufficient to give the location.

Figure 2. It is obvious but could you add an x-axis label (time/years)

**Authors comments;** Yes we can.

Figure 5. could directional arrows be placed on the photograph e.g. orientation photograph was taken from or is looking towards.

**Authors comments;** Thanks for the excellent suggestion. We can amend the caption to link in more with Fig. 1;

"Figure 5. Images *(looking southwest) from eastern timelapse camera location (Fig. 1)* of calving activity at the glacier terminus in summer 2017."

Dr Jenna Sutherland (Leeds Beckett University)

**References**

Truffer, M. and Motyka, R.J., 2016. Where glaciers meet water: Subaqueous melt and its relevance to glaciers in various settings. *Reviews of Geophysics*, *54*(1), pp.220-239.

---

## Author Comment (AC2)

Review of Dye et al., Warm proglacial lake temperatures and thermal undercutting drives rapid retreat of an Arctic glacier.

**Reviewer 1 comments (all in bold)**

**This submission presents a collection of field- and satellite-derived datasets to characterise terminus fluctuations, surface elevation changes, and calving patterns/mechanisms at a proglacial Arctic lake. The primary conclusion is that frontal ablation is a key component of glacier mass loss, and that undercutting from warm lake temperatures is a key driver of rapid terminus retreat. These are logical interpretations to make, and are largely uncontroversial, but they are only weakly substantiated by the evidence that is presented, leaving the reader to take a large amount on trust. This is probably the most significant of a number of major issues with the manuscript in its current form. These are noted below, and given I believe there will have to be quite a bit of re-writing before this work could be published, I have stopped short of highlighting typographic and editorial errors at this stage.**

**Authors comments:** Thank you for the very detailed and carefully constructed review that you have given. We appreciate the detailed comments about the manuscript that represents the first substantial study of Kaskasapakte, which presents numerous challenges in how to carefully synthesise 3 fieldwork campaigns at a remote and relatively complex glacier-lake system. Your comments are very much appreciated in addressing the challenges of producing a well rounded holistic article that captures as many elements of the system as possible, without becoming too complex and overbearing.

1. **There is, at present, no clear overriding research question that this study seems to focus on addressing. A well-designed study identifies a gap in knowledge, formulates a data collection strategy to shed light on that gap, collects those data, and then analyses the results to provide an advance in knowledge.**

   **Authors comments;** We agree in an ideal study where aspects of the system are previously known/proven then selected knowledge gaps can be identified and studied in an empirical fashion. The lack of previous study at the field site demands a holistic approach, in order to assess the inter-relations of the different components of the system where possible and identify knowledge gaps requiring further study to advance the science further.We will include the following aim statement on line 51;

   'This study aims to investigate the coupling between glacial lake warming and glacier retreat, (on a decadal and seasonal timescale) at a previously unstudied Arctic glacier through remote sensing and in situ measurements.'

2. **The data presented here seek to address four key objectives that are loosely connected but that each demand a major effort by themselves to reach some substantial and novel conclusions (i.e. they could and maybe should each be a study in their own right). As a result, the effort is spread too thinly, and each of the datasets are deficient in some way, undermining the interpretation;**

**Authors comments;** We appreciate the reviewer has identified the challenges of synthesising data from different components at a remote and highly complex glacial-lake system. This is necessitated by the lack of prior study at this field site, due to the remoteness rather than the lack of dynamic processes. We feel that presenting the data that we have available is highly valuable in progressing the science of glacier and lake interactions and will thoroughly review and amend the discussion of limitations. We feel that it is better to keep the objectives specified as they are by particular methodologies.

a. **for example, if lake thermal characteristics are to be revealed then it really needs more than a single sensor to say something robust,**

**Authors comments;** We appreciate the reviewer has identified the challenges of combining complex limnology data in association with glaciology data. We also present lake temperatures of 3.5 $^{o}$C from 20 m depth next to the glacier terminus from 8$^{th}$ August 2019 – whilst this is only a very limited snapshot it does prove the lake to be well mixed during the 2019. We also have further temperature data (3 to 4 $^{o}$C) to support this from the central portion of the lake; this is included in the supplementary (we are happy to make this more explicit in the manuscript). We also have lake temperature from 2017 (3 $^{o}$C at the ice front) and 2022 (3 $^{o}$C at the ice front) – we are open to including this data but felt this may upset the balance between synthesis of information and an already complicated narrative (as the reviewer has kindly identified).

b. **and similarly if calving rates are to be defined then 2D camera imagery acquired over six weeks of the melt season is not really sufficient.**

**Authors comments;** We appreciate the reviewer has identified timespan of the timelapse imagery data. We have made no claim or comments regarding defining calving rates at Kaskasapakte from this data in this manuscript. We have future plans for analysing this data in 3D.

c. **This leaves the authors with a rather speculative discussion about what may or may not be driving mass loss at this site and making some quite significant leaps between observations and interpretations, which leaves the reader unconvinced that they are at all robust.**

**Authors comments;** We have reported a synthesis of what we have observed and recorded at Kaskasapakte glacier to lake system over three field seasons. Thank you for

identifying (comments above) that these observations are logical and largely uncontroversial. We acknowledge acquiring further data to fully understand the system would be fundamental to fully understand the interactions, so would like to amend the title as we accept, we cannot attribute a sole driver to the retreat of the glacier. However, we feel that the manuscript still presents an important advance in the study of glacier-lake interactions.

Dye et al. (2022) identified Kaskasapakte to have the highest retreat rate in the surrounding area. Our manuscript builds on this analysis by analysing retreat rates at higher temporal and spatial resolution, between 2008 and 2019. We have shown that lake temperatures have been warmer (above 4 C) than has been assumed and processes associated with thermal erosional undercutting (as proven with sonar) have been highly prevalent during fieldwork in 2017 and 2019. We are happy to include data from Landsat surface temperature retrieval (Ermida et al., 2020) that has no recordings of temperature above 4 C prior to 2000 BCE. Furthermore, frequent calving activity has been classified and mainly associated with thermo-erosional undercutting during field work periods. We have quantified geometric/volumetric changes in the lower portion of the glacier (we were unable to climb any further up loose moraine/mountainside to obtain further images for the SfM model).

We also wish to request for an extension to map occurrence and extent of iceberg cover in the proglacial lake in the past satellite record. As well as further investigating the velocity products/data that you have kindly highlighted.

Whilst we acknowledge that we have been unable to measure all components of this glacial system, we have provided a relatively thorough analysis of the overall system within the logistical limits of a small budget study at a remote field site location. The evidence that we present is not fully comprehensive of the entire system, but we feel that it is imperative to publish in order that better resourced studies can test the theory more conclusively. With this in mind we would like to amend the title to;

"Warm proglacial lake temperatures and thermal undercutting *enhances* rapid retreat of an Arctic glacier"

3. **Perhaps as a consequence of the above, there is no common thread that can be navigated through the different sections. Different datasets are introduced at different times, describing different aspects of analysis over different timescales. It is very difficult to keep track of what is going on at each step, and it requires a lot of checking backwards and forwards to remind oneself what has already been introduced and what its purpose was. In a similar way the discussion jumps from one aspect to another, with many elements benefitting from only a single paragraph comprising several sentences on the findings of other studies rather than making deep and insightful interpretations of the results that have been presented here. The manuscript would benefit from a re-design to tell a coherent story about the specific problem or question**

**to be solved; at present it unfortunately fails to do this in any meaningful way.**

**Authors comments;** We accept that the narrative is complex and challenging; a reflection of the complex and challenging system that it reports on. We will further review the manuscript and include greater organisation by timescale of each component (inter-annual, seasonal and event) as well as reviewing the intertwinement of the glacial and lake system.

Given the highly dynamic nature of such glacier-lake systems it is imperative that studies into them are published to further scientific understanding and also identify key knowledge gaps that enable these systems to be fully understood. We feel the manuscript provides a firm base for future studies into the glacier-lake system at Kaskasapakte (and other field sites) to be studied in order to facilitate greater understanding of how they have evolved and also predict future responses to climate. Given the speed of Arctic Amplification of climate change, we feel that it is imperative to publish such work in good time.

4. **The presentation of the methods is currently quite difficult to navigate.**

**Authors comments;** Thank you for your thoughts on this, we will review this section and try to develop a schematic or figure to locate more methods from the field campaign – Figure S1 in the supplementary goes some way to achieving this. We will investigate amending Figure 1 c to include more Methodological information , although a graphical abstract may prove to be a better visual tool to guide the reader.

**a. I have read this section several times and still don't follow certain elements (in particular what metres of recession per melt year is – lines 87-92 are really confusing.**

**Authors comments;** Thank you for your thoughts on this, we will review this section and revise this section (potentially with an equation to simplify the different terms). We will refer to metres of recession per year and include further equations in the Methods to explain how it is calculated.

**b. and how subaqueous mass loss was calculated without a corresponding 2015 bathymetric dataset?).**

**Authors comments;** Thank you for your thoughts on this, we assume a vertical subaqueous ice front in 2015; which is likely given the lack of debris cover to insulate an ice foot protruding into the lake (as has been reported from debris covered glaciers in New Zealand and the Himalaya). Given the lack of debris cover we assume no buried ice to be present in the lake bed – as it would have been subject to buoyancy and calved. So we assume the lake bathymetry from surveys in 2022 to be inherited from retreat since 2015. We will amend this section to clarify this assumption by changing line 115 to;

"The Volume Calculation tool in QGIS was used to calculate the volume changes between the 2019 and 2015 DEMs down to lake level *(with a vertical subaqueous ice front assumed in both years).*"

**c. I also find several aspects to be missing – for example different lake temperature measurements at different depths pop up in the manuscript discussion and in the SI that are not mentioned at all within the methods section.**

**Authors comments;** Thank you for identifying the oversight in not mentioning data presented in the Supplementary in the Methods section – we shall amend this by amending line 133 to;

*'Further temperature measurements (also HOBO UAA-002-08) were taken at 20m depth on 8th August 2019 for 3 hours and at the central position in the lake (see S1) at 5m deep from 5th August to 9th September; both results are presented in the supplementary section.'*

5. **A major deficiency is the almost complete absence of any uncertainty estimates – uncertainty on the DEM calculations (associated with any offsets or biases between the two datasets, which come from very different sources/methods – were these co-registered in any way? And what sort of values were acquired in stable off-ice areas), uncertainty on the sonar depths, uncertainty on the measurement of the ice-cliff positions and the height of the ice face (that is then used in calculations). This need calculating, and adding, to every presented quantitative value.**

   - **Authors comments;** Thank you for highlighting this substantial oversight. We will review and include uncertainty calculations for both DEMs as well as investigating co-registration. Include this in descriptions of ice cliff heights
   - Include plumb line measurements that were taken in 2019 to validate the sonar depths (thankfully one of the co-authours remembered these)
   - Review and include uncertainty values for the ice front positions

6. **The calculation of mass loss is fundamentally flawed by the absence of any ice dynamics information. As the authors note themselves, terminus positions are a composite effect of the forward motion of the glacier and the removal of mass by melt and calving processes. And surface elevation changes are a composite effect of any vertical**

**component of the ice velocity (emergence in this case), dynamic effects, and surface melt/sublimation.**

**Authors comments;** Thank you for your thoughts on this, please note that we do not purport to calculate mass loss. We report elevation changes and acknowledge the limitations that you mention within the text. Unfortunately we were unable to conduct a surface mass balance survey due to the limited size (3 to 4 people) and restrained logistics (field camp) of our field campaigns. Whilst we have no ice density or compaction data, we will investigate whether data from neighbouring glaciers may help and consider whether this would represent a feasibly robust approach for calculating mass loss.

a. **To dismiss ice velocity as being negligible because other glaciers in the area are slow flowing is not acceptable. Velocity data are now widely available, for example the velocities from Kaskasapakte Glacier are readily available from Millan et al., 2022, and a quick look suggests these are not negligible, as stated in the manuscript (Ref: https://doi.org/10.1038/s41561-021-00885-z).**

**Authors comments;** Thank you for your thoughts on this and identifying this potential dataset. We will remove the assumption that velocity rates are low based on surrounding glaciers. Given the lack of debris cover on the glacier body (there is problematic debris at the margins), potential for geolocation problems at 68 °N (projection and image registration) and relative course spatial resolution of the data used in the suggested velocity product; we would like to request an extension to investigate the accuracy of it on surrounding glaciers where velocity fields are known. It may be possible to use selected data inputs to the product, correct geolocation errors and reduce noise within the dataset to provide a useful velocity record.

b. **It would not be a big step to incorporate these measurements into the calculations and make them a lot more robust. An additional observation here though is that the % contributions of each form of mass loss are highly dependent on the areal extent of the surface elevation change analysis – this seems to be an arbitrary distance from the terminus at present, whereas to be able to talk about mass loss from the system this needs to be integrated across the entire glacier. Otherwise, the information collected is simply surface elevation change, not mass loss.**

**Authors comments;** Thank you for your thoughts on this, as mentioned above we have not claimed to report mass loss (although I appreciate the original title infers this). Unfortunately we could not travel any further during the SfM survey due to loose unconsolidated moraine and steep mountainsides; so the limit of the SfM model is limited by the extent of relatively safe ground travel. We have reported surface

elevation change and suggested what needs to be achieved at this glacier to improve future understanding and how mass loss will respond. We accept that the original title did incorporate this claim into it though and have since proposed a modified title (which removes the single driver assumption). We are investigating other data that may be used for deriving elevation data.

7. **A final more general point is that to be able to convince the reader that one thing is driving another, it is necessary to show that the effect you have observed has happened because of some behavioural aspect of the control.**

   **Authors comments;** Thank you for the very pertinent comment. We feel that changing the title from 'drives' to 'enhances' retreat deals with the limitation of the study that you have rightly identified. We would also be happy to include lake surface temperature data from Landsat (Ermida et al., 2020) that shows no recordings of 4 C prior to 2000 BCE.

   a. **Here, the lake has been in existence for multiple decades, maybe even a century, and will have warmed up every summer and cooled down every winter, albeit with some warming over the long-term. So what is it that has changed recently to cause the rapid terminus recession?**

      **Authors comments;** We note that the lake has existed since the end of the Little Ice Age in the study area. Further retreat details (since 1959) are in Dye et al., (2022), which we refer to but will interweave into the Introduction more. Unfortunately the only data on elevation for the upper part of the glacier are for 2015, which has limited elevation change analysis to the lower part of the glacier where the SfM survey was focused. We accept that whilst this is a substantial limitation – surface lowering of the lower section of the glacier reaches a maximum of 7 to 8 m between 2015 and 2019 – roughly 2m per year. This is incorporated into the volume calculations and attributed as such in the manuscript. The revised title is more in acknowledgement of the limitation that surface lowering in the accumulation area has not been accounted for.

   b. **There is a hint in the discussion that calving may have increased, but without evidence.**

      **Author comments;** We believe that this is a reasonable postulation given the data we present. We also have images of the glacier from pre 2008, that show a lack of an active calving front (the ice front smoothly grades down to the waterline), which we are happy to include in the supplementary as further evidence. We would also like to request an extension to investigate mapping iceberg appearance/persistence in the proglacial lake from the satellite imagery record. We appreciate that these are limited snapshots in time though, so we are prepared to remove this assertion.

c. **There is also some suggestion that recent heatwaves may have contributed, but again there is no long-term weather station data presented to show this. Is it not much more likely that the glacier is responding to a negative climate forcing, and the ice flux has reduced, and that has caused the rapid recession?**

**Author comments;** We refer to Dye et al. (2021) for detailed analysis of weather station data and also Dye et al. (2022) for comparison to retreat rates of other land terminating glaciers. Kasakasapakte has retreated substantially faster than other land terminating glaciers in the region (see Dye et al., 2022). We shall also include some further statistics from this on line 70;

*"Recently the area has experienced pronounced heatwaves (month long), with August 2014 and July 2018 being 5.4 oC and 5.6 oC above the long term average (Dye et al., 2022)."*

d. **Unless some change in the forcing can be shown, and/or all other possibilities can be discounted, the conclusion of a single or key driver being responsible for the changes is highly suspect, especially when this is the same driver that has been around for many years or decades beforehand.**

**Authors comments;** We acknowledge this limitation, which is also acknowledged in the text and are only able to comment on the relative balance of processes within the system. The title will be amended to reflect this. We would also like to request further time to investigate and present other data sources to quantify different components of the system.

**I do think there are some valuable observations within the data that are presented here, and with some careful thinking about (and reformulation of) the manuscript structure they will be worthy of publication. Unfortunately I do not support the publication of this submission in its current form.**

**Authors comments;** We would like to thank the reviewer for the very carefully considered and philosophically deep review of the paper. The comments will be very useful in further amending the manuscript and we feel that publication of which will provide a very useful contribution into current understanding of glacier-lake interactions, as well as providing further impetus and directing further studies and publications in this field. Further to this and the proposed changes suggested above, we would like to request an extension to investigate including other sources of data to provide a more comprehensive analysis of the system.

---

## Author Response (AR1)

**Warm proglacial lake temperatures and thermal undercutting enhances rapid retreat of an Arctic glacier**

Dear Editor and Reviewers,

Thank you for your careful reviews of the article and carefully considered suggestions for improvements. We have tried to incorporate these suggestions for improvement as far as possible, whilst trying to strike the balance between the broad range of datasets (given the lack of previous studies) whilst also trying to simplify the narrative and improve the navigability of the article. Given the paucity of previous process based studies from lacustrine terminating Arctic glaciers, we feel that an observation based article, such as this, is important to publish. As it provides temperature data directly from the ice front and a characterisation of the glacier-lake system; thus providing an important basis to inform future studies for further hypothesis development and closer constraining of the interlinkages in the system. This is essential for understanding how the system will respond to future warming from Arctic Amplification.

Given the diverse nature of the methods used to build an overview of a complex and dynamic system, we have chosen to maintain the methodological based objectives and improved signposting to improve the navigability between these different sections for the reader. We have also provided a graphical abstract to map out the different components of the study for the reader. We feel that although the amalgamation of different methodologies at different timescales is perhaps non-optimal, the lack of previous studies at such sites in the Arctic necessitates this approach to build up understanding of such dynamic systems in a rapidly changing part of the climate system.

Best regards,

The authorship team.

---

## Author Response (AR2)

**Kas Glacier Paper; Second Review**

Reviewer 1 comments;

General comments
This study provides a characterization of an understudied system, combining a variety of observational perspectives. In my estimation, some of the major concerns raised by the reviewers and editor have been addressed, while others still need some revision. I believe that the presented data are definitely worth publishing and provide valuable insight into the recent (last decade) development of KG and KGL.

Thank you for your kind and carefully considered review, which we feel has been a great help in further improving the manuscript.

However, the stated aim of the study stated in the research question (L53) and objective 4 (L64) does not quite match what the data and analysis accomplish. Lake warming – which is presented as a central point in the research question – is only inferred rather than directly or indirectly measured. There is however evidence for comparatively high lake temperatures, which in my opinion is justification enough to publish. Consequently, I believe the framing of the study needs to be slightly adjusted to fit the data at hand. In my view, the strong points of the study, supported by a lot of observations, are objectives 2 and 3, which should receive more emphasis.

We have changed 'warming' to 'temperatures' in line 53;

This study aims to investigate the relationship between glacial lake temperatures  and glacier retreat, (on a decadal and seasonal timescale) at a previously unstudied Arctic glacier through remote sensing and in situ measurements.

We have refrained from re-numbering the objectives as they are numbered to reflect the current flow of the paper (rather than by importance) – as each subsection is currently ordered to follow the objective order. We could letter the objectives a,b,c,d if it removes the suggestion of ordering by importance that numbering gives?

Similarly, one might argue that a regressive lake bed topography (which is not well-constrained in the area of recent retreat) could lead to accelerated retreat rather than warming temperatures or a combination of both. Generally, the data on lake temperatures are quite limited – in particular spatially. It is my opinion that this data should not be overused to infer lake properties. Instead, it should be recognized as a limitation more explicitly.

Thank you for raising these important points. We have further temperature data showing very similar near surface temperatures (~3.5 to 4°C) from a series of boat surveys; which we are happy to make available on request (such as in a data repository or in the Supplementary). We feel that the thermistor data presented here is from important strategic positions; next to the ice front at 1m depth (note; North facing so

remained in shadow), next to the ice front at ~20m depth (where cold dense meltwater may have been expected) and at the central point of the lake at 5m depth. We feel that these strategic sampling points provide a fairly sound representation of the lake characteristics during 2019; particularly as other locations in the lake were more prone to iceberg disruption (such as near the outlet as experienced in 2017).

We argue that there was little evidence of mechanical calving in the 2017 or 2019 field season, as crevassing was inconsistent with either extensional flow or buoyant forces. Given the thinning of the terminus (and likely cold based ice) it seems logical to argue that mechanically driven calving during earlier periods was less likely, so depth would likely have lesser importance on calving mechanisms. Although we can't conclusively eliminate the possibility of a shallower area in the bathymetry data gap, so we have made the following amendments;

**Line 177**

The bathymetry survey revealed that central parts of the terminus remained in relatively deep water during the survey period (2008 to 2019) but note the gap in sonar bathymetry data in this area (grey area; Fig.2).

**Line 307**

Unfortunately, a gap (~50 m across) in the sonar survey of KGL remains, but we suggest that the gap bathymetry is likely to be similar, as it is too small to have a substantial over-deepening and there is no obvious evidence for any substantial shallowing (such as a bedrock riegel or moraine) as large icebergs passed to the lake mid-point.

**Line 426**

Whilst variability in mechanical support due to lake depth cannot be conclusively eliminated (due to data gap in the bathymetry) we argue that mechanical calving processes are not the main driver of recession of KG terminus, due to the factors discussed above and factors further discussed in the next section.

Finally, in my experience reading the paper, it remains quite challenging to keep track of the different datasets collected at various points in time. Perhaps the publication would profit from a data collection overview in the form of a table and/or timeline.
Thanks for the excellent suggestion. We have added one at the start of the Methods (Table 1).

Specific comments
1. The bathymetric survey is also spatially limited and some subsequent characterizations of the lake bed (L175-177; L306-310) are likely a result of the interpolation using these sparse measurement points. It is unfortunate that the areas without (or with only very limited) bathymetric data are exactly where the terminus was more stable and then started retreating rapidly from. In my view, this should also be acknowledged more clearly as a point of uncertainty.
Thank you for the insightful review comments. We have revised the following sections accordingly;

Section 4.1

"whereas the eastern margin is likely to deepen (>5m) steeply within 10m of the shore (note sparser depth points on E margin; see Supplementary S1)."

Section 5.1

"Unfortunately, a gap (~50 m across) in the sonar survey of KGL remains, but we suggest that the gap bathymetry is likely to be similar, as it is too small to have a substantial over-deepening and there is no obvious evidence for any substantial shallowing (such as a bedrock riegel or moraine) as large icebergs passed to the lake mid-point."

Section 5.4.2 2019 Field Season

Whilst variability in mechanical support due to lake depth cannot be conclusively eliminated (due to data gap in the bathymetry) we argue that mechanical calving processes are not the main driver of recession of KG terminus, due to the factors discussed above and factors further discussed in the next section.

Section 5.5 – Wider Implications

"Comprehensive sonar surveys of the ice proximal lake bed are essential to identify any possible shallower areas that may have provided sufficient support to maintain terminus position stability."

**2. The observations of the calving front are well-supported by data – though some interpretations seem to rely on time-lapse footage. Maybe these videos could be included as a supplement.**

We have included the timelapse video in a Zenodo repository (included in Assets);
https://zenodo.org/records/16631366?token=eyJhbGciOiJIUzUxMiJ9.eyJpZCI6IjI0M2M2
NDAzLTEzMWUtNDUxZS1hYWNlLTNjOTBlYjkwOTBmNSIsImRhdGEiOnt9LCJyYW5kb20i
OiIwM2VhYTkxMTQyZTZlYWI0NTk5N2Q5MTBlNzZmNjhkNiJ9.KsC_4akPRzK_sykp4_mgk
JqGVQxWTG5o7TYJIqKKXeSp0SqUTBfo7S46Ki7ojVniNaHqMjNM4Yp8AAZziNLvew

https://doi.org/10.5281/zenodo.16631365

**3. Debris-covered areas of the glaciers appear at some point in the discussion, they could quickly be mentioned in the study area description.**

Thanks for the suggestion, we have added the following (in blue) to the Study Area section;

Kaskasapakte Glacier (KG) is ~2 km long and flows northeast from two subsidiary corries (located below ~500 m headwalls, with peaks of ~2,000 m asl to the east, south and west) into the main trunk; currently terminating in a calving front in an unnamed proglacial lake (now referred to as KGL) with some latitudinal supraglacial debris bands near the terminus (Fig. 1 and see Supplementary for images).

**4. I assume that light intensity (L267, Fig. 7) is used as an indication of whether the thermistor was moved from its intended position, maybe this could be shortly explained somewhere in the methods.**

We have amended the following section;

**3.4 Lake water and meteorological changes through the 2019 melt season**

Summer 2019 lake water temperatures were measured at 1 m depth on a line suspended from the glacier terminus (67.95396 N 18.55955 E) using a HOBO UAA-002-08 pendant to measure hourly temperature (± 0.5 °C) and light (so periods of solar warming from sensor disruption could be identified for quality control).

5. In section 5.2, the addition of ITS_LIVE velocities makes sense to get a rough estimate of ice dynamics. Still, I think the two concepts of (local) mass loss (which is well-constrained from your data) and mass balance (where the dynamics are an unconstrained component) need to be separated clearly. As it is worded now, "volume of ice lost" (L323) could refer to either of these quantities.

Thank you for raising this important point and apologies for the inconsistencies that were present. We have included a sentence at the start of section 3.3.1 to clarify and have made changes to consistently refer to 'static volume' change;

"We calculate 'static' glacier volume change for geometric differences between glacier surfaces in the lower part of KG, as insufficient velocity data prevented dynamic processes being incorporated."

Section 4.2

Ice surface velocities derived from NASA ITS_LIVE feature tracking were below ~ 40 m y$^{-1}$ (see S6; Gardner et al., 2019) but high uncertainties prevented incorporation into geometric calculations; so presented volume changes are essentially 'static' as dynamics were not incorporated.

Section 5.2

There has also been substantial surface lowering of KG between 2015 and 2018, with changes of 7 to 8 m extending 200 m back up glacier across the front, with 774,374 m$^3$ reduction in glacier volume  in the 2019  terminus area between 2015 and 2019 (RMSE = 0.52 m; Fig. 4).

6. One of the processes only touched on briefly is water entering the lake from sub-/englacial meltwater channels. To me, what happens with incoming meltwater seems quite relevant to the thermal state of the lake. Was there any indication that a plume formed, bringing cold water to the surface? Or would you expect sediment-rich (at least temporarily denser) meltwater to remain at the bottom of the lake? Is this significant to the circulation in the lake?

Excellent points. I have added this to the start of section 4.4. Calving Mechanisms;

"Buoyant plumes were not observed at the ice front."

At the time of observations in 2019 the lake was relatively well mixed, with temperatures from thermistor 1, 2 (central) and 3 (terminus deep) all between 3.5 to 4°C. We also have near surface temperature readings from boat surveys between 3.5 to 4°C and would be happy to share this data. This was following a period of north easterly winds blowing down the lake, which likely mixed the lake given it's relatively shallow depth. We agree that lake thermal stratification can have a strong influence on the subaqueous glacier profile, as has been reported from Patagonia (Sugiyama et al., 2016). Given the relatively simple lake thermal stratification observed in 2019, we have refrained from discussing how Kas Glacier lake thermal stratification may vary over time in this paper, as we feel that would be best left for a more detailed process based paper. We have made an addition in section 5.5 Wider Implications to highlight this;

This study provides an observational overview of the system and future research should focus on the evolution of subaerial and subaqueous lacustrine glacier termini (in conjunction with lake thermal stratification),

particularly given future predictions of air temperature increases and associated reduction of lake ice cover duration, which will expose lake water to greater warming influences (Huang et al., 2022).

7. L417/418: the longitudinal supraglacial debris ridges at the margins do not look like thrust debris (of subglacial origin) to me, which is the type of debris cover the sources you cite refer to. Rather I believe the debris to be of supraglacial origin – through rockfalls and avalanches – as it is common on debris-covered glaciers. You can even visually trace the debris transport up-glacier to individual rock walls. I would not use these features as clear evidence for cold ice at the margins.

Thank you for your excellent points regarding the supraglacial debris and we apologise that we had removed an image from the supplementary that showed latitudinal debris bands more clearly. I have included an image of the terminus from 1988, which has clearer debris bands extending laterally across the terminus. These are similar to those cited visually, but we are unable to confirm whether the debris is subglacial or supraglacial. Given the lack of radar data from this glacier we would like to include this evidence, whilst it is relatively weak we feel it helps to pose an important point as to whether cold based ice from the terminus has been removed by calving.

We have amended the second paragraph in section 5.4.3 Mechanical Calving to;

Unfortunately there is no existing radar survey to confirm the thermal structure of KG. However, we argue that the margins and terminus are likely to be frozen to the bed (as at neighbouring Storglaciaren) as water filled crevasses persisted through the melt season and the supraglacial debris bands seen near the terminus of KG (see Supplementary 6) are typically associated with the transition of temperate to cold based ice (Moore et al., 2011; Monz et al., 2022; Carrivick et al., 2023).

8. The legends, captions, and axis labels in the supplement could be improved:
- In S1, one part of the legend is in the figure and another part as text in the caption
Thanks for identifying this, we have now labelled the ground control points on the figure as we feel this is consistent with the other labelling.
- In Table 1, it is not clear which observations the data originate from (probably time-lapse?)
Ah. Apologies for the oversight. Now amended to include timelapse information. REPOSITORY?
- In S4, axis labels are missing altogether
Thanks. Amended.
- In S5, three of the five subfigures are labelled a-c, which is not reflected in the caption

Technical corrections
L159-162: Somehow two sources of precipitation data at the same location are mentioned here (SMHI and SITES).
Thanks for the observation – there was a change in operator of the AWS.
L173/174: this sentence does not quite add up to me.

Thanks. We have tried to clarify this sentence further by adding "only" and "; as they are";

Whilst these satellite images represent only a snapshot in time, we report them here as there is noticeable variation between 2008 and 2019 that should be considered in the context of lacustrine terminus geometry changes; as they are over time periods that are important to the typical progression of thermal undercutting by a proglacial lake.

L187: I think this heading is a bit misleading, as the data presented in this section only represent a single state of the terminus rather than change

Excellent point – 'change' deleted. Thanks.

Fig. 4: not sure if this is the best colour scale – to me the green areas intuitively represent mass gain.

We experimented with different colour schemes but felt this was the best available one in QGIS and feel the legend clarifies that there is no increase in volume.

L216: This sentence is still a bit confusing and should be formulated more clearly.

Ah sorry for that! Thanks. We have reworded it to;

"Removal of the 2015 glacier ice surface down to the 2019 lake level, varied between 0 to 23 m in height across the pink polygon area (Fig.4)."

L217: there is no Figure 4a, probably Fig. 4 is meant here.

Thanks. Amended to Fig.4

L221/222: are these 30.2% calculated based on the entire ablation area or just the black outline from Fig. 4?

Thanks for identifying this – we have deleted '' and now refer to 'terminus area (black and pink polygon)' instead.

Fig. 7: the colours in subfigure d) do not quite match the legend and c).

Apologies for this, it is due to the mismatch between the colour scheme from Python and from the graphics software. We feel that they are distinguishable still.

L320: unit consistency; use either m y-1, m yr-1, or m a-1 throughout the entire text.

We have now eliminated m a$^{-1}$ and ensured m y$^{-1}$ is used consistently.

L359/374: these headings could be more descriptive, they currently look more like placeholders.

We have changed these to;

5.4.1 2017 Field Season Changes in Glacier Front

5.4.2 2019 Field Season Changes in Glacier Front and Environmental Conditions

L452/453: this sentence does not quite add up to me.

Thanks. We've reworded it to;

This study provides the first (to our knowledge) direct field evidence of  Arctic glacier recession being enhanced by contact with warm (>4 °C) proglacial lake temperatures, through rapid thermo-erosional undercutting and associated calving.

Reviewer 2 comments;

The authors have revised their manuscript in line with reviewer suggestions and I appreciate the authors detailed responses and explanations. I am satisfied my comments have been carefully considered and the authors responses are thoroughly justified. It is clear to the reader what the study aims to achieve and the methods and datasets are now more transparent. As such, I think the manuscript is much improved, making a significant contribution to glacier lake observations and providing important data with which to refine and validate glacier evolution models, for example. The minor comments below are just personal preferences but not at all detrimental to the paper if they are not acted upon.

Thank you for the carefully considered review.

Lines 68-70 – 'Two periods of fieldwork were conducted (between 23rd July-4th August 2017 and 29th July-10th August 2019) at an unnamed proglacial lake (now referred to as KGL) in the Kebnekaise massif (Arctic Sweden), into which Kaskasapakte Glacier (KG) terminates. Further sonar bathymetry surveys conducted in August 2022 are also included.' I think these sentences are better suited to the beginning of the Methods section and an alternative introduction to the Study area.

Thanks – I've moved these to the Methods section (which has improved the start of that section too).

L94 – insert 'Swedish' before Lantmateriet?

Done. Thanks.

L136 - Changes in surface elevation of KG between 2015-2019 'was calculated' by subtracting…?

Thanks. Added!

L155 – 'Arches are defined as features where depth is less than height of the overhang, whereas caves are where depth is more than the height of the overhang.' Sorry, I'm not sure I entirely understand this sentence, depth of what?

Apologies for the confusion here. I've hopefully clarified that sentence a bit further to;

"For classification of larger undercut features in the subaerial ice front (above the waterline notch); arches are defined as features where depth of the overhang was less than its height , whereas  if the depth of the overhang was greater than its height it was defined as a cave."

L182 – is it a shallower 'bedrock' shelf (?)

It wasn't possible to identify whether this shallower shelf was predominantly composed of moraine or bedrock. I've just added '(bedrock/moraine composition unknown)'.

L264 – 'Water temperatures were monitored in the ice proximal area (at 1 m depth; Fig.7) from 29th July to 28th August 2019' – Would it be useful to report the average temperature in brackets?

Yes! Excellent suggestion. Added.

L284 – Fig 7 caption – '1 m depth next to the glacier terminus' – it's a little unclear if the thermistor string was parallel to the glacier terminus, or if it's a single point?

Ah! Thanks for spotting this and sorry for the confusion – I've tried to iron out the inconsistencies and added the following sentence to section 3.4;

"Summer 2019 lake water temperatures were measured at 1 m depth on a line suspended from the glacier terminus… A thermistor was positioned lower down (at 2m depth) parallel to the ice front, but this was removed by a calving event."

L450 – 'observed in order to understand what changes in balances at the ice front that may trigger periods of frequent calving activity.' – Suggest deleting 'that'

Thanks. Done

It might be useful to add the temperature measurement locations as points on Fig 1 to provide some context.

Yes, I see what you mean. It wouldn't work on Fig.1 as it's from 2008. Unfortunately we couldn't get hold of a high quality satellite image from 2019 to plot the thermistors at the relevant positions – it's very shady!

I've added them to Figure 2 (and referred to it in the text) – I think that should work and it's important to see the lake depth context for each thermistor I think.

---

## Author Response (AR3)

**Egusphere-2024-2510; Response to Editors comments**

Thank you for the kind and considerate reviews and comments. We made these changes;

Table 1 -> Met data-> "Meteorological data" Done
Ln 158-> instead of overhang, would "overhang depth" or "undercutting distance" be better? Changed to "overhang depth"
Ln 413 -> I was a bit confused by "variability in mechanical support." Can this be better defined? Is it spatial or temporal, for instance? Changed to;

"Whilst variability in lake depth affecting calving rates cannot be conclusively eliminated between ~ 2012 to 2016 (due to data gap in the bathymetry) we argue that mechanical calving processes are not the main driver of recession of KG terminus, as indicated by the lack of crevassing (either from buoyancy or extensional flow) (Tsutaki et al., 2013)."

Supplement S6-> Please define in the caption the yellow dashed line and the red circle in the photo. Done